# Beyond Distribution Estimation: Simplex Anchored Structural Inference Towards Universal Semi-Supervised Learning

Yaxin Hou [1]   Jun Ma [1]   Hanyang Li [1]   Bo Han [1]   Jie Yu [2]   Yuheng Jia [1 3]

## Abstract

Semi-supervised learning faces significant challenges in realistic scenarios where labeled data is scarce and unlabeled data follows unknown, arbitrary distributions. We formalize this critical yet under-explored paradigm as Universal Semi-supervised Learning (UniSSL). Existing methods typically leverage unlabeled data via pseudo-labeling. However, they often rely on the idealized assumption of a uniform unlabeled data distribution or require sufficient labeled data to estimate it. In the UniSSL setting, such dependencies lead to numerous erroneous pseudo-labels, thereby triggering representation confusion. Fortunately, we observe that inter-sample relations captured by representations are more reliable than pseudo-labels. Leveraging this insight, we shift our focus to representation-level structural inference to bypass distribution estimation. Accordingly, we propose Simplex Anchored Graph-state Equipartition (SAGE), which captures high-order inter-sample dependencies to establish structural consensus for guiding representation learning. Meanwhile, to mitigate representation confusion, we employ vectors that satisfy a simplex equiangular tight frame to serve as a coordinate frame for guiding inter-class representation separation. Finally, we introduce a weighting strategy based on distribution-agnostic metrics to prioritize reliable pseudo-labels and an auxiliary branch to isolate potentially erroneous pseudo-labels. Evaluations on five standard benchmarks show that SAGE consistently outperforms state-of-the-art methods, with an average accuracy gain of **8.52%**.

[1]School of Computer Science and Engineering, Southeast University, Nanjing, China. [2]School of Electrical Engineering, Southeast University, Nanjing, China. [3]Key Laboratory of New Generation Artificial Intelligence Technology and Its Interdisciplinary Applications (Southeast University), Ministry of Education, Nanjing, China. Correspondence to: Yuheng Jia <yhjia@seu.edu.cn>.

*Proceedings of the $43^{rd}$ International Conference on Machine Learning*, Seoul, South Korea. PMLR 306, 2026. Copyright 2026 by the author(s).

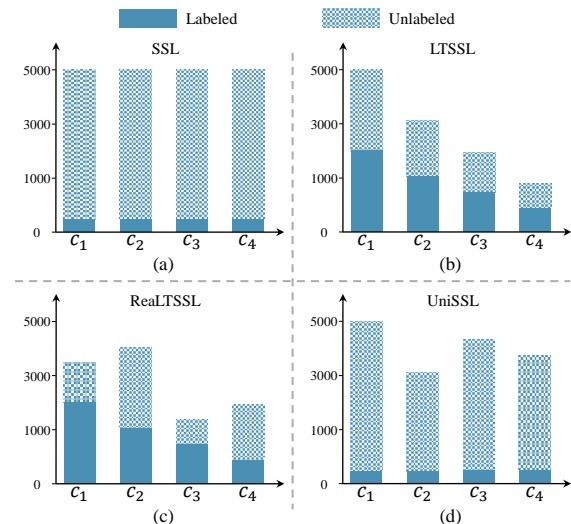

*Figure 1.* Comparison of class distributions across semi-supervised learning (SSL), long-tailed semi-supervised learning (LTSSL), realistic long-tailed semi-supervised learning (ReaLTSSL), and the proposed universal semi-supervised learning (UniSSL) settings. UniSSL tackles a more challenging and realistic scenario characterized by extremely scarce labeled data and unknown, arbitrary unlabeled data distributions.

## 1. Introduction

Semi-supervised learning (SSL) has emerged as a critical paradigm for training deep neural networks (DNNs) in label-scarce domains, such as medical diagnosis (Wang et al., 2025; Huang et al., 2025). By utilizing the model's high-confidence predictions to generate pseudo-labels for unlabeled samples, SSL effectively leverages large-scale unlabeled data to supplement limited supervision. However, traditional SSL frameworks (Wang et al., 2023; Cheng et al., 2025) typically rely on the idealized assumption of consistent and uniform class distributions between labeled and unlabeled data (Fig. 1(a)). While long-tailed semi-supervised learning (LTSSL) (Lee et al., 2021; Fan et al., 2022) has been developed to address class-imbalanced distributions, it often assumes that the unlabeled data distribution mirrors the long-tailed nature of the labeled data (Fig. 1(b)). More recently, realistic long-tailed semi-supervised learning (ReaLTSSL) (Du et al., 2024; Hou et al., 2025) has been introduced to tackle scenarios where unlabeled data

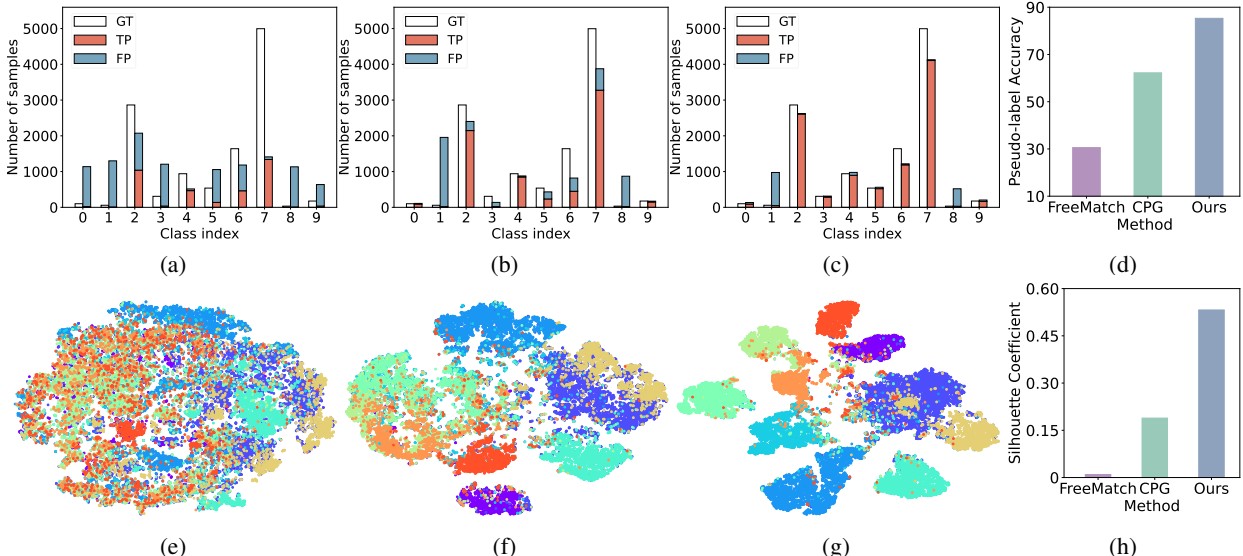

*Figure 2.* Impact of pseudo-label quality on representation learning. **(a)–(c)** Class distributions of pseudo-labels generated by FreeMatch (Wang et al., 2023), CPG (Hou et al., 2025), and our SAGE under an arbitrary unlabeled data distribution. GT denotes the ground-truth distribution, while TP and FP represent true positive and false positive pseudo-labels, respectively. **(d)** Comparison of pseudo-label accuracy (%) among the three methods. **(e)–(g)** t-SNE visualizations of learned representations on the test set for FreeMatch, CPG, and our SAGE, respectively. **(h)** Quantitative evaluation of representation quality via the silhouette coefficient ($\uparrow$). All experiments are conducted on SVHN with $(N_{max}, M_{max}, \gamma_l, \gamma_u) = (4, 4996, 1, 150)$, where $N_{max}$ ($M_{max}$) denotes the sample count of the most frequent labeled (unlabeled) class, and $\gamma_l$ ($\gamma_u$) is the corresponding imbalance ratio. The results show that low-quality pseudo-labels trigger representation confusion in existing methods. In contrast, our SAGE establishes structural consensus via high-order inter-sample dependencies to bypass distribution estimation, and employs a simplex equiangular tight frame to guide representation separation. This enhances representation discriminability, which in turn improves pseudo-label quality.

follows unknown, arbitrary distributions (Fig. 1(c)), which is a more practical setting, since the class distribution of unlabeled data is typically inaccessible in the real world. Nevertheless, the performance of current ReaLTSSL methods remains constrained under extreme label scarcity, as the lack of supervision impedes reliable estimation of such distributions, thereby hindering the generation of high-quality pseudo-labels. This combination of extreme label scarcity and distributional uncertainty poses significant challenges that have yet to be fully addressed.

In response to these challenges, we formalize the paradigm of Universal Semi-supervised Learning (UniSSL), which targets unknown, arbitrary unlabeled data distributions under extreme label scarcity (Fig. 1(d)). Existing efforts to tackle unlabeled data can be broadly categorized into two paradigms, yet both fail under UniSSL. First, standard SSL methods (e.g., FreeMatch (Wang et al., 2023)) typically rely on an idealized uniform distribution assumption of the unlabeled data. By imposing distribution alignment or entropy maximization, they introduce an implicit class-balance prior. Under the arbitrary distributions inherent to UniSSL, this prior forces the model to generate pseudo-labels that falsely conform to a uniform distribution, leading to massive false positives (Fig. 2(a)). Second, recent ReaLTSSL methods (e.g., CPG (Hou et al., 2025)) attempt to dynamically esti-

mate the unlabeled data distribution. However, they suffer from an estimation breakdown under UniSSL's extreme label scarcity. Without sufficient supervision, the estimated distribution becomes highly unstable, further compromising pseudo-label quality (Figs. 2(b) and 2(d)). Together, these failures trigger representation confusion, as evidenced by qualitative t-SNE visualizations (Figs. 2(e) and 2(f)) and a sharp decline in the silhouette coefficient, which measures the ratio of inter-class separation to intra-class compactness (Fig. 2(h)). Such representation confusion demonstrates how biased distribution priors and poor initial supervision can mislead the model, ultimately destroying the discriminability of learned representations.

The fundamental bottleneck of existing SSL and ReaLTSSL methods within UniSSL lies in their reliance on unreliable pseudo-labels. In Fig. 3, we conduct a diagnostic study to evaluate whether inter-sample relations can correct incorrect pseudo-labels during training. Specifically, we measure the correction ratio, defined as the percentage of unlabeled samples whose initially incorrect pseudo-labels are successfully rectified by the inter-sample relations. **We observe that this ratio increases steadily as training progresses and eventually stabilizes at a high level, indicating that inter-sample relations consistently capture more reliable semantic structure than pseudo-labels.** These results em-

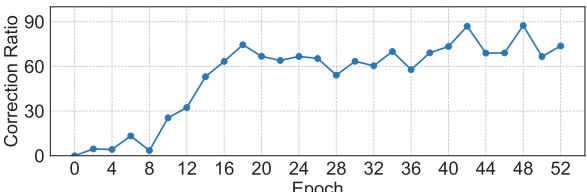

*Figure 3.* Effectiveness of inter-sample relations in pseudo-label rectification. These results empirically validate that inter-sample relations provide a more accurate and robust supervisory signal than pseudo-labels.

pirically confirm that inter-sample relations provide a more accurate and robust supervisory signal than pseudo-labels in UniSSL.

Motivated by these insights, we propose Simplex Anchored Graph-state Equipartition (SAGE), a framework that shifts the paradigm from distribution estimation to structural inference. At its core, SAGE focuses on the representation level to establish structural consensus by capturing high-order inter-sample dependencies across the data manifold. Specifically, we develop a Graph-state Relational Inference (GRI) module that utilizes multi-step graph transitions to provide stable structural guidance, effectively bypassing the reliance on unreliable pseudo-labels. To further prevent representation confusion, SAGE employs vectors satisfying a simplex equiangular tight frame as fixed geometric anchors. These anchors serve as a stable coordinate frame for guiding class-specific representations toward distinct directions, thereby enhancing inter-class representation separation (Figs. 2(g) and 2(h)). Finally, to ensure the quality of supervision, we introduce Distribution-agnostic Reliability Prioritization (DRP) to prioritize reliable pseudo-labels using distribution-agnostic metrics (i.e., max-confidence and top-two margin). To further isolate potentially erroneous signals, we incorporate an auxiliary branch that decouples the processing of unlabeled data from the primary classification head. Collectively, these mechanisms establish a virtuous cycle where structural consensus enhances representation discriminability, which in turn improves pseudo-label quality, preventing representation confusion. Comprehensive experiments on five commonly used benchmarks (i.e., CIFAR-10, CIFAR-100, Food-101, SVHN, and STL-10) under various scenarios validate that SAGE achieves new state-of-the-art performance with an average accuracy improvement of **8.52%**.

In summary, our **contributions** are as follows:

1. We formalize Universal Semi-supervised Learning (UniSSL), a more realistic paradigm targeting unknown, arbitrary unlabeled data distributions under extreme label scarcity, and identify that the reliance on unreliable pseudo-labels leads to representation confusion.

2. We observe that inter-sample relations are more reliable

than pseudo-labels. Therefore, we propose SAGE to shift the focus from distribution estimation to representation-level structural inference. We introduce a Graph-state Relational Inference module that captures high-order dependencies to establish a stable structural consensus for guiding the model.

3. We employ a simplex equiangular tight frame as fixed geometric anchors to guide inter-class representation separation. Furthermore, we develop a Distribution-agnostic Reliability Prioritization mechanism and an auxiliary branch to effectively prioritize reliable pseudo-labels and isolate potentially erroneous ones.

## 2. Related Work

**Semi-supervised learning** (SSL) has emerged as a critical paradigm for mitigating label scarcity by leveraging extensive unlabeled data to enhance model generalization. Prevailing SSL methods are primarily built upon two core strategies: consistency regularization (Miyato et al., 2019; Ke et al., 2019), which enforces prediction invariance across varied perturbations (e.g., weak and strong augmentations), and pseudo-labeling (Chen et al., 2018; Cascante-Bonilla et al., 2021; Han et al., 2026), which utilizes high-confidence model predictions on unlabeled data to guide self-training. This integration is exemplified by FixMatch (Sohn et al., 2020), which aligns predictions between weak and strong augmentation views of the same sample. Building upon this architecture, subsequent studies such as FlexMatch (Zhang et al., 2021), FreeMatch (Wang et al., 2023), SoftMatch (Chen et al., 2023), SemiReward (Li et al., 2024), and CGMatch (Cheng et al., 2025) have introduced various thresholding strategies to better balance the quality and quantity of pseudo-labels. However, these methods typically rely on the idealized assumption of consistent and uniform distributions between labeled and unlabeled data, a prior that is often operationalized through distribution alignment or entropy maximization. In the context of UniSSL, where unlabeled data follows unknown, arbitrary distributions, this distributional mismatch introduces significant bias into pseudo-label generation, severely compromising the model's generalization in realistic applications.

**Long-tailed semi-supervised learning** (LTSSL) addresses the challenges of class imbalance to extend SSL efficacy to real-world scenarios. Traditional LTSSL methods typically assume that the unlabeled data distribution mirrors that of the labeled data, augmenting standard SSL frameworks (e.g., FixMatch (Sohn et al., 2020)) with established long-tailed learning techniques (Zhang et al., 2023; Hou et al., 2024; Li et al., 2025; Li & Jia, 2025; Li et al., 2026; Jia et al., 2024) to mitigate bias towards majority classes. Representative strategies include re-sampling or selective pseudo-labeling (Peng et al., 2025; Hou et al., 2025), feature enhancement (Fan et al., 2022), and re-weighting consis-

tency losses (Lai et al., 2022). More recently, attention has shifted towards realistic long-tailed semi-supervised learning (ReaLTSSL), which accounts for potential class distribution mismatches between labeled and unlabeled data. Methods in this domain employ adaptive logit adjustment (Wei & Gan, 2023; Du et al., 2024) or multi-expert ensemble frameworks (Ma et al., 2024; Hou & Jia, 2025) to capture diverse unlabeled data distributions. However, a critical gap remains in their practical applicability: many methods (e.g., ACR (Wei & Gan, 2023), CPE (Ma et al., 2024), and Meta-Expert (Hou & Jia, 2025)) require prior knowledge of the unlabeled data distribution, which is often unavailable. Conversely, while methods like SimPro (Du et al., 2024) and CPG (Hou et al., 2025) attempt to explicitly or implicitly estimate the unlabeled data distribution dynamically, their estimation accuracy degrades significantly under extreme label scarcity inherent in UniSSL (Fig. 2(b)). This failure to accurately capture unknown, arbitrary distributions leads to biased pseudo-labeling, thereby triggering representation confusion (Fig. 2(f)).

Additional related work is detailed in Appendix A.

## 3. Method

### 3.1. Problem Formulation and Challenges

In UniSSL, we consider a labeled dataset $\mathcal{D}_l = \{x_i^l, y_i^l\}_{i=1}^N$ of size $N$ and an unlabeled dataset $\mathcal{D}_u = \{x_j^u\}_{j=1}^M$ of size $M$, where both datasets share identical representation and label spaces. Here, $x_i^l$ denotes the $i$-th labeled sample with ground-truth label $y_i^l \in \{1, \ldots, C\}$, $x_j^u$ represents the $j$-th unlabeled sample, and $C$ is the total number of classes. Let $N_c$ denote the number of labeled samples in class $c$, and define the labeled data imbalance ratio as $\gamma_l = \frac{\max_c N_c}{\min_c N_c}$. While one can theoretically define $M_c$ as the number of samples for class $c$ in $\mathcal{D}_u$ and $\gamma_u = \frac{\max_c M_c}{\min_c M_c}$ as the unlabeled data imbalance ratio, in practice, the class distribution of unlabeled data is unknown and arbitrary, making $M_c$ and $\gamma_u$ inaccessible. The goal of UniSSL is to train a model $F : \mathbb{R}^d \to \{1, \ldots, C\}$ (i.e., a backbone with a classification head), parameterized by $\theta$, using $\mathcal{D}_l$ and $\mathcal{D}_u$ to achieve robust generalization. Specifically, we focus on a highly challenging scenario characterized by extreme label scarcity (e.g., minimal labels per class) and unknown, arbitrary unlabeled data distributions.

### 3.2. Proposed Framework

**Graph-state relational inference.** Leveraging the insight that inter-sample relations are more reliable than pseudo-labels (as evidenced in Fig. 3), we propose Graph-state Relational Inference (GRI) to shift the focus from pseudo-labeling to structural inference. Specifically, GRI captures high-order inter-sample dependencies by mining the data

manifold spanned by unlabeled samples and the set of fixed geometric anchors $\mathbf{P} = [\mathbf{p}_1, \ldots, \mathbf{p}_K]^\top \in \mathbb{R}^{K \times d}$. These anchors, whose rigorous geometric construction is detailed in the next subsection, provide a stable coordinate frame. This allows for the formulation of inter-sample relations as a structural consensus that guides the refinement of individual representations.

Given the projection head output $\mathbf{z}_i \in \mathbb{R}^d$ for an unlabeled sample, we derive a relational embedding $\mathbf{a}_i \in \mathbb{R}^K$, which serves as a geometric signature that encodes the sample's position within the rigid coordinate frame. This is formulated as a reconstruction optimization problem:

$$\min_{\mathbf{a}_i} \|\mathbf{z}_i - \mathbf{a}_i \mathbf{P}\|_2^2 + \lambda \|\mathbf{a}_i\|_2^2, \tag{1}$$

where $\| \cdot \|_2^2$ denotes the squared $L_2$ norm and $\lambda > 0$ is the regularization parameter that promotes a dense relational representation. The closed-form solution is given by:

$$\mathbf{a}_i = (\mathbf{z}_i \mathbf{P}^\top)(\mathbf{P}\mathbf{P}^\top + \lambda \mathbf{I})^{-1}, \tag{2}$$

where $\mathbf{I}$ denotes the identity matrix.

Building upon these signatures, we structure the data manifold via the structural affinity matrix $\mathbf{A}$, where $\mathbf{A}_{ij} = \langle \mathbf{a}_i, \mathbf{a}_j \rangle$. To capture high-order inter-sample dependencies, we define a row-normalized state transition matrix $\hat{\mathbf{P}} = \text{Softmax}(\mathbf{A})$ and derive the structural consensus $\mathbf{G}$ via $\beta$-step graph transitions as $\mathbf{G} = \hat{\mathbf{P}}^\beta$. This diffusion process propagates relational information throughout the data manifold, providing more stable structural guidance. We then introduce the structural contrastive loss $\mathcal{L}_{con}$ to align the instance-wise similarity $\mathbf{S}$ (where $\mathbf{S}_{ij} = \text{Sigmoid}(\langle \mathbf{z}_i, \mathbf{z}_j \rangle)$) with the structural consensus $\mathbf{G}$:

$$\mathcal{L}_{con} = \text{BCE}(\mathbf{S}, \text{sg}[\mathbf{G}]), \tag{3}$$

where $\text{sg}[\cdot]$ denotes the stop-gradient operation. By propagating relational information, the structural consensus $\mathbf{G}$ captures a collective agreement on the geometry of the data manifold. This allows the model to rectify erroneous pseudo-labels if they contradict the structural consensus (as evidenced in Fig. 3), thereby providing a stable training signal even when the pseudo-labels are unreliable.

Finally, to further stabilize the representation, we introduce a complementary regularization term $\mathcal{L}_{sim}$. While $\mathcal{L}_{con}$ focuses on mining high-order inter-sample dependencies, $\mathcal{L}_{sim}$ ensures the local smoothness of the representation space by maximizing the similarity between backbone features $\mathbf{f}$ and projected features $\mathbf{z}$ across different augmentation views:

$$\mathcal{L}_{sim} = -\mathbb{E}[\text{sim}(\mathbf{z}_i^w, \text{sg}[\mathbf{f}_i^s]) + \text{sim}(\mathbf{z}_i^s, \text{sg}[\mathbf{f}_i^w])], \tag{4}$$

where $\text{sim}(\cdot, \cdot)$ denotes cosine similarity. These components establish a learning paradigm that is label-agnostic, moving beyond the reliance on distribution estimation.

**Simplex equiangular anchor generation.** As established in the GRI module, the integrity of structural inference relies on a stable coordinate frame. Furthermore, to mitigate the representation confusion identified in Sec. 1, it is essential to provide a geometric structure that guides inter-class representation separation. The simplex equiangular tight frame naturally fulfills these dual requirements, as it provides a set of fixed vectors that are both maximally and equiangularly separated in the representation space.

To realize this construction, we first obtain an orthogonal matrix $\mathbf{Q} \in \mathbb{R}^{d \times d}$ via QR decomposition of a random Gaussian matrix to define the initial orientation of the anchors. To ensure the anchors are zero-centered, we employ the centering matrix $\mathbf{O} = \mathbf{I}_K - \frac{1}{K}\mathbf{1}_K\mathbf{1}_K^\top \in \mathbb{R}^{K \times K}$, where $K = d+1$. Let $\mathbf{V} \in \mathbb{R}^{K \times d}$ be the matrix whose columns are the $d$ orthonormal eigenvectors of $\mathbf{O}$ corresponding to its non-zero eigenvalues. By applying the orthogonal transformation $\mathbf{Q}$ to the basis $\mathbf{V}$ and incorporating the necessary scaling factor, the final anchor matrix $\mathbf{P} = [\mathbf{p}_1, \ldots, \mathbf{p}_K]^\top \in \mathbb{R}^{K \times d}$ is computed as:

$$\mathbf{P} = \sqrt{\frac{K}{K-1}}\mathbf{V}\mathbf{Q}^\top. \tag{5}$$

This closed-form construction guarantees that the anchors are perfectly centered at the origin, i.e., $\mathbf{P}^\top\mathbf{1}_K = \mathbf{0}$. Each anchor $\mathbf{p}_i$ (i.e., the $i$-th row of $\mathbf{P}$) is unit-norm and satisfies the equiangular property $\mathbf{p}_i^\top\mathbf{p}_j = -\frac{1}{K-1}$ for all $i \neq j$, providing the maximal equiangular separation for $K$ vectors in $\mathbb{R}^d$. The key advantage of these anchors lies in their distribution-agnostic nature, as the anchors in SAGE are fixed simplex equiangular anchors generated once before training rather than learnable prototypes. By maintaining invariant geometric relations regardless of the sample count in each class, they provide the rigid coordinate frame necessary to ground the structural inference process in the GRI module. Consequently, SAGE effectively decouples representation learning from unstable distribution priors, ensuring inherent robustness to the unknown, arbitrary unlabeled data distributions in UniSSL, as evidenced by the highly discriminative representation space visualized in Fig. 2(g).

**Distribution-agnostic reliability prioritization.** To maximize unlabeled data utilization under distributional uncertainty, we introduce Distribution-agnostic Reliability Prioritization (DRP). DRP evaluates the reliability of pseudo-labels through distribution-agnostic metrics and dynamically prioritizes the reliable ones.

Specifically, for each unlabeled sample, we evaluate the reliability of its pseudo-label through two metrics derived from the weak augmentation view prediction $\mathbf{q}_w$: (i) the max-confidence $q_{max} = \max(\mathbf{q}_w)$, which captures the model's absolute certainty, and (ii) the top-two margin $q_{gap} = q_w^{(1)} - q_w^{(2)}$, where $q_w^{(1)}$ and $q_w^{(2)}$ denote the high-

est and second-highest class prediction probabilities, respectively. $q_{max}$ reflects the absolute prediction confidence, whereas $q_{gap}$ serves as a measure of relative discriminability, focusing on separation.

To adapt to the evolving state of the model, we maintain exponential moving averages (EMAs) of the means ($\mu$) and variances ($\sigma^2$) for both metrics throughout the training process. Based on these statistics, we assign a weight $w \in (0, 1]$ to each unlabeled sample using a truncated Gaussian kernel. For a specific metric $q_\kappa$, its corresponding weight contribution is formulated as:

$$\mathcal{W}(q_\kappa; \mu_\kappa, \sigma_\kappa) = \exp\left(-\frac{[\min(0, q_\kappa - \mu_\kappa)]^2}{2\sigma_\kappa^2}\right), \tag{6}$$

where $\exp(\cdot)$ denotes the exponential function and $\kappa \in \{max, gap\}$ denotes the metric type. This formulation ensures that samples performing above the moving average mean $\mu_\kappa$ receive a full weight of 1.0, while those below are exponentially penalized based on their distance from the mean $\mu_\kappa$. The final adaptive weight is computed as $w = \mathcal{W}(q_{max}; \mu_{max}, \sigma_{max}) \cdot \mathcal{W}(q_{gap}; \mu_{gap}, \sigma_{gap})$. By leveraging these measures, DRP prioritizes reliable pseudo-labels in a distribution-agnostic manner, thereby providing a stable supervisory signal that complements the structural consensus of GRI.

**Auxiliary branch.** To further mitigate the impact of unreliable pseudo-labels, we decouple the learning process by introducing an auxiliary classification head $\phi_{aux}$. During training, the primary classification head $\phi_{cls}$ is optimized exclusively on labeled data to preserve the purity of the task-specific decision boundary:

$$\mathcal{L}_{cls} = \mathbf{H}(\phi_{cls}(\mathbf{f}^l), y^l), \tag{7}$$

where $\mathbf{H}(\cdot)$ denotes the cross-entropy loss. In contrast, $\phi_{aux}$ processes both labeled and unlabeled data, where the pseudo-labels for the auxiliary classifier are generated by the primary classifier, capitalizing on its superior decision boundary purity. This architectural isolation ensures that gradients induced by potentially noisy pseudo-labels are primarily confined to the auxiliary parameters, effectively safeguarding the primary classification head from direct exposure to unreliable supervision. The adaptive weight $w$ derived from the DRP module is then applied to the consistency loss:

$$\mathcal{L}_{aux} = w \cdot \mathbf{H}(\phi_{aux}(\mathbf{f}_s^u), \mathrm{sg}[\hat{y}_w^u]) + \mathcal{L}_{aux}^{sup}, \tag{8}$$

where $\hat{y}_w^u$ is the aforementioned pseudo-label and $\mathcal{L}_{aux}^{sup}$ denotes the supervised loss for labeled data computed by the auxiliary classification head. By integrating dual-head isolation with distribution-agnostic weighting, this auxiliary branch serves as a second line of defense that complements the structural consensus established by GRI, ensuring the model's robustness under extreme label scarcity and distributional uncertainty.

## 3.3. Model Training and Prediction

In the training phase, we first generate the fixed simplex equiangular anchors. The model is then trained end-to-end by jointly optimizing the primary classification head, the auxiliary classification head, and the graph-state relational inference module. The overall objective function $\mathcal{L}_{total}$ is defined as the sum of four loss components:

$$\mathcal{L}_{total} = \mathcal{L}_{cls} + \mathcal{L}_{con} + \mathcal{L}_{sim} + \mathcal{L}_{aux}, \qquad (9)$$

where $\mathcal{L}_{cls}$ is the supervised loss derived from the primary classification head, $\mathcal{L}_{con}$ and $\mathcal{L}_{sim}$ are the structural contrastive and representation consistency losses from the graph-state relational inference module, and $\mathcal{L}_{aux}$ is the loss from the auxiliary classification head, which encompasses both the weighted consistency loss on unlabeled data and the supervised loss on labeled data. The training procedure is summarized in Algorithm 1 in Appendix B. Upon completion of training, the auxiliary classification head and the projection head are discarded. For an unseen test sample $x^*$, the predicted label $y^*$ is obtained by selecting the class index with the maximum logit from the primary classification head $\phi_{cls}$:

$$y^* = \mathrm{argmax}_{c \in \{1, \ldots, C\}} \phi_{cls}(\mathbf{f}^*)_c. \qquad (10)$$

## 4. Experiments

### 4.1. Experimental Setting

**Datasets.** We conduct our experiments on five widely-used datasets (i.e., CIFAR-10 (Krizhevsky, 2009), CIFAR-100 (Krizhevsky, 2009), Food-101 (Bossard et al., 2014), SVHN (Netzer et al., 2011), and STL-10 (Coates et al., 2011)), following the main experimental settings in FreeMatch (Wang et al., 2023) and CPG (Hou et al., 2025), with details provided in Appendix C.

**Baselines.** We compare our SAGE with a comprehensive set of state-of-the-art methods. These include four SSL algorithms (i.e., FixMatch (Sohn et al., 2020), FreeMatch (Wang et al., 2023), SoftMatch (Chen et al., 2023), and CG-Match (Cheng et al., 2025)) and five LTSSL algorithms (i.e., ACR (Wei & Gan, 2023), SimPro (Du et al., 2024), CD-MAD (Lee & Kim, 2024), Meta-Expert (Hou & Jia, 2025), and CPG (Hou et al., 2025)). Moreover, we use the supervised learning (SL) setting as a performance upper-bound reference. For a fair comparison, we test these baselines and our SAGE on the widely-used codebase USB[1]. For the data augmentation strategy, an identical weak augmentation is applied to both labeled and unlabeled data, while strong augmentation is reserved for unlabeled data. We use the same dataset splits with no overlap between labeled and unlabeled training data for all datasets.

### 4.2. Implementation Details

We follow the default settings and hyperparameters in USB, i.e., the batch size of labeled data $B_l$ is set to 64, while that of unlabeled data $B_u$ is set to 7 times $B_l$. We resize all input images to $32 \times 32$ for all datasets. Moreover, we use the WRN-28-2 (Zagoruyko & Komodakis, 2016) architecture, the SGD optimizer with momentum 0.9, and weight decay 5e-4 for training. We use a cosine learning rate decay (Loshchilov & Hutter, 2017) scheme, which sets the learning rate to $\eta \cos\left(\frac{7\pi t}{16T}\right)$, where the initial learning rate $\eta$ is set to 0.03, $t$ is the current training step, and the total number of training steps $T$ is set to $2^{18}$. The regularization parameter $\lambda$ is set to 0.1 and the number of graph transition steps $\beta$ is set to 5. **All these hyperparameters are fixed, and our method is not sensitive to their specific settings. A detailed parameter sensitivity analysis is provided in Appendix D.** We repeat each experiment with three different random seeds and report the mean and standard deviation of the results. We conduct the experiments on a single NVIDIA RTX 4090 GPU using PyTorch v2.1.0. Our code is made available[2].

### 4.3. Main Result

The results for CIFAR-10 and CIFAR-100 are summarized in Tables 1 and 2, while the results for SVHN, Food-101, and STL-10 are presented in Table 3.

**Performance in universal SSL scenarios.** The superiority of our SAGE is most pronounced in the proposed UniSSL paradigm, where unlabeled data follows unknown, arbitrary distributions. On CIFAR-10 with ($N = 40, M = 11650$, arbitrary), SAGE achieves 61.24% accuracy, exceeding the recent CPG (50.24%) by **11.00** percentage points (pp). This performance advantage extends to more complex benchmarks. For instance, on SVHN, our method achieves a remarkable accuracy of 84.11% under the arbitrary setting, yielding a substantial improvement of over **24** pp over existing approaches. Such consistent improvements across diverse benchmarks stem from our graph-state relational inference (GRI) module.

**Performance in standard SSL scenarios.** In scenarios where unlabeled data follows a uniform distribution, SAGE also demonstrates a clear competitive advantage. As shown in Table 1, SAGE attains an accuracy of 80.68% under the ($N = 40, M = 13980$) setting, surpassing the leading baseline CGMatch (72.83%) by **7.85** pp. This trend remains consistent on CIFAR-100, where SAGE achieves an accuracy of 37.82% under the ($N = 400, M = 17100$) setting, outperforming CPG by 3.71 pp. These results highlight that even in the absence of distribution mismatch, existing methods remain vulnerable to unreliable pseudo-labels

---

[1] https://github.com/microsoft/Semi-supervised-learning

[2] https://github.com/Yaxin-ML/SAGE

*Table 1.* Comparison of accuracy (%) on CIFAR-10 under the $\gamma_l = \gamma_u$ and $\gamma_l \neq \gamma_u$ settings. For this dataset, we set $\gamma_l = 1$ and $\gamma_u \in \{1, 50, 100, 150\}$. CE and LA denote using softmax cross-entropy loss and logit-adjusted softmax cross-entropy loss under the supervised learning (SL) setting, respectively. We use **bold** to mark the best results.

| Scenario and Method | | $N = 40$, $M = 13980$ | | | $N = 40$, $M = 12390$ | | | $N = 40$, $M = 11650$ | | | Average Accuracy |
| --- | --- | --- | --- | --- | --- | --- | --- | --- | --- | --- | --- |
| | | Uniform $\gamma_u = 1$ | Long-tailed $\gamma_u = 50$ | Arbitrary $\gamma_u = 50$ | Uniform $\gamma_u = 1$ | Long-tailed $\gamma_u = 100$ | Arbitrary $\gamma_u = 100$ | Uniform $\gamma_u = 1$ | Long-tailed $\gamma_u = 150$ | Arbitrary $\gamma_u = 150$ | |
| SL | CE | 91.23 ±0.22 | 84.24 ±0.35 | 82.98 ±3.82 | 90.77 ±0.14 | 79.49 ±0.64 | 79.00 ±4.53 | 90.52 ±0.06 | 75.83 ±0.40 | 76.01 ±5.25 | 83.34 ±1.71 |
| | LA (ICLR'21) | 91.31 ±0.19 | 88.13 ±0.19 | 87.53 ±1.19 | 90.79 ±0.17 | 85.79 ±0.40 | 85.18 ±1.86 | 90.47 ±0.16 | 84.22 ±0.37 | 83.15 ±2.22 | 87.40 ±0.75 |
| SSL | FixMatch (NeurIPS'20) | 58.56 ±2.03 | 54.23 ±5.48 | 56.94 ±5.35 | 61.26 ±12.27 | 44.14 ±5.93 | 51.39 ±10.81 | 58.79 ±1.85 | 45.25 ±4.65 | 48.36 ±5.29 | 53.21 ±5.96 |
| | FreeMatch (ICLR'23) | 72.23 ±2.58 | 50.58 ±8.76 | 57.66 ±9.08 | 71.91 ±1.58 | 47.49 ±1.69 | 50.41 ±7.30 | 69.94 ±10.16 | 46.85 ±6.33 | 45.38 ±1.84 | 56.94 ±5.48 |
| | SoftMatch (ICLR'23) | 72.37 ±2.93 | 53.68 ±3.71 | 54.33 ±11.79 | 72.00 ±1.80 | 46.69 ±1.79 | 45.21 ±3.60 | 65.44 ±3.86 | 48.36 ±7.05 | 42.82 ±4.25 | 55.66 ±4.53 |
| | CGMatch (CVPR'25) | 72.83 ±2.88 | 55.45 ±7.56 | 59.57 ±11.64 | 65.55 ±9.03 | 51.34 ±2.79 | 51.54 ±10.39 | 70.36 ±1.74 | 48.82 ±4.99 | 49.92 ±3.46 | 58.38 ±6.05 |
| LTSSL | ACR (CVPR'23) | 61.12 ±6.80 | 51.02 ±0.36 | 58.82 ±4.25 | 55.40 ±1.94 | 40.01 ±2.57 | 56.36 ±4.38 | 57.52 ±3.19 | 43.37 ±1.90 | 45.06 ±10.18 | 52.08 ±3.95 |
| | SimPro (ICML'24) | 16.07 ±2.07 | 16.40 ±0.60 | 15.80 ±0.78 | 16.91 ±1.65 | 16.86 ±2.97 | 16.16 ±2.01 | 15.86 ±1.54 | 16.71 ±0.94 | 15.89 ±0.29 | 16.29 ±1.43 |
| | CDMAD (CVPR'24) | 69.21 ±8.13 | 42.39 ±5.52 | 50.95 ±5.00 | 63.01 ±1.38 | 42.01 ±2.07 | 45.57 ±3.38 | 63.15 ±2.62 | 41.31 ±4.87 | 44.04 ±4.04 | 51.29 ±4.11 |
| | Meta-Expert (ICML'25) | 48.75 ±3.69 | 37.64 ±5.43 | 45.37 ±3.20 | 46.76 ±2.39 | 36.71 ±3.47 | 42.72 ±4.36 | 46.38 ±1.79 | 34.15 ±3.41 | 41.45 ±5.21 | 42.22 ±3.66 |
| | CPG (NeurIPS'25) | 65.10 ±1.05 | 50.59 ±4.66 | 52.82 ±6.41 | 62.40 ±1.31 | 49.75 ±6.07 | 52.41 ±2.58 | 62.16 ±5.67 | 49.05 ±4.02 | 50.24 ±8.01 | 54.95 ±4.42 |
| | SAGE (Ours) | **80.68** ±2.42 | **68.11** ±4.99 | **69.43** ±8.19 | **77.82** ±3.83 | **62.42** ±2.70 | **65.99** ±3.72 | **76.96** ±4.95 | **59.26** ±2.61 | **61.24** ±4.72 | **69.10** ±4.24 |

*Table 2.* Comparison of accuracy (%) on CIFAR-100 under the $\gamma_l = \gamma_u$ and $\gamma_l \neq \gamma_u$ settings. For this dataset, we set $\gamma_l = 1$ and $\gamma_u \in \{1, 5, 10, 15\}$. We use **bold** to mark the best results.

| Scenario and Method | | $N = 400$, $M = 24600$ | | | $N = 400$, $M = 19400$ | | | $N = 400$, $M = 17100$ | | | Average Accuracy |
| --- | --- | --- | --- | --- | --- | --- | --- | --- | --- | --- | --- |
| | | Uniform $\gamma_u = 1$ | Long-tailed $\gamma_u = 5$ | Arbitrary $\gamma_u = 5$ | Uniform $\gamma_u = 1$ | Long-tailed $\gamma_u = 10$ | Arbitrary $\gamma_u = 10$ | Uniform $\gamma_u = 1$ | Long-tailed $\gamma_u = 15$ | Arbitrary $\gamma_u = 15$ | |
| SL | CE | 70.93 ±0.07 | 69.38 ±0.31 | 68.76 ±0.70 | 68.48 ±0.50 | 64.84 ±0.33 | 64.10 ±0.79 | 67.02 ±0.31 | 62.17 ±0.55 | 61.63 ±0.80 | 66.37 ±0.48 |
| | LA (ICLR'21) | 70.83 ±0.14 | 70.18 ±0.22 | 70.08 ±0.25 | 68.40 ±0.30 | 66.39 ±0.30 | 66.39 ±0.67 | 67.34 ±0.50 | 64.36 ±0.16 | 64.46 ±0.61 | 67.60 ±0.35 |
| SSL | FixMatch (NeurIPS'20) | 37.57 ±3.85 | 38.04 ±2.47 | 37.11 ±2.58 | 35.30 ±3.86 | 34.32 ±1.82 | 34.60 ±3.05 | 33.01 ±3.89 | 31.02 ±2.03 | 32.05 ±1.62 | 34.78 ±2.80 |
| | FreeMatch (ICLR'23) | 36.53 ±1.70 | 32.88 ±3.29 | 33.59 ±1.76 | 30.13 ±0.97 | 27.22 ±2.67 | 28.30 ±2.09 | 28.35 ±2.84 | 24.76 ±2.51 | 25.72 ±2.22 | 29.72 ±2.23 |
| | SoftMatch (ICLR'23) | 36.20 ±2.24 | 34.32 ±1.52 | 33.45 ±2.14 | 29.90 ±2.67 | 27.96 ±2.80 | 28.06 ±2.36 | 28.22 ±1.27 | 25.62 ±2.00 | 26.22 ±1.65 | 29.99 ±2.07 |
| | CGMatch (CVPR'25) | 39.44 ±1.75 | 36.64 ±1.84 | 36.66 ±1.71 | 33.90 ±2.42 | 29.66 ±3.96 | 31.41 ±2.92 | 31.32 ±3.36 | 27.60 ±3.83 | 30.00 ±3.08 | 32.96 ±2.76 |
| LTSSL | ACR (CVPR'23) | 38.81 ±3.45 | 38.53 ±2.04 | 37.91 ±2.02 | 35.54 ±2.44 | 34.08 ±2.44 | 35.19 ±0.83 | 33.93 ±2.55 | 31.01 ±2.90 | 31.75 ±2.05 | 35.19 ±2.30 |
| | SimPro (ICML'24) | 19.10 ±1.78 | 18.62 ±1.96 | 18.10 ±0.38 | 19.74 ±0.27 | 19.02 ±1.54 | 16.74 ±0.63 | 18.66 ±1.99 | 17.97 ±1.04 | 18.21 ±1.31 | 18.46 ±1.21 |
| | CDMAD (CVPR'24) | 26.35 ±0.76 | 27.12 ±3.74 | 25.92 ±1.06 | 24.90 ±5.65 | 23.07 ±1.36 | 21.86 ±1.96 | 21.36 ±2.58 | 20.53 ±1.48 | 19.37 ±1.19 | 23.39 ±2.20 |
| | Meta-Expert (ICML'25) | 29.68 ±1.76 | 26.53 ±2.69 | 28.70 ±0.61 | 26.31 ±0.81 | 21.82 ±2.15 | 24.13 ±0.74 | 24.79 ±0.20 | 21.17 ±1.55 | 22.94 ±1.42 | 25.12 ±1.32 |
| | CPG (NeurIPS'25) | 38.09 ±2.78 | 37.22 ±3.89 | 37.57 ±3.09 | 37.19 ±2.03 | 34.55 ±2.17 | 34.41 ±2.68 | 34.11 ±1.15 | 31.86 ±2.38 | 32.48 ±2.22 | 35.28 ±2.49 |
| | SAGE (Ours) | **41.00** ±2.21 | **39.31** ±1.97 | **40.11** ±0.80 | **38.76** ±1.70 | **36.31** ±1.65 | **36.58** ±0.64 | **37.82** ±1.31 | **34.18** ±1.68 | **34.25** ±0.66 | **37.59** ±1.40 |

*Table 3.* Comparison of accuracy (%) on SVHN, Food-101, and STL-10 under the $\gamma_l = \gamma_u$ and $\gamma_l \neq \gamma_u$ settings. We set $\gamma_l = 1$ for all datasets, with $\gamma_u \in \{1, 150\}$ for SVHN and $\gamma_u \in \{1, 15\}$ for Food-101. We use **bold** to mark the best results. $N/A$ denotes the unknown $\gamma_u$ in STL-10 since ground-truth labels for the unlabeled dataset are inaccessible.

| Scenario and Method | SVHN | | | Food-101 | | | STL-10 | Average Accuracy |
| --- | --- | --- | --- | --- | --- | --- | --- | --- |
| | $N = 40$, $M = 11650$ | | | $N = 404$, $M = 17271$ | | | $N = 40$, $M = 100k$ | |
| | Uniform $\gamma_u = 1$ | Long-tailed $\gamma_u = 150$ | Arbitrary $\gamma_u = 150$ | Uniform $\gamma_u = 1$ | Long-tailed $\gamma_u = 15$ | Arbitrary $\gamma_u = 15$ | $N/A$ $\gamma_u = N/A$ | |
| FixMatch (NeurIPS'20) | 72.27 ±15.60 | 60.62 ±11.37 | 58.74 ±14.08 | 10.43 ±0.75 | 9.30 ±0.36 | 9.82 ±0.59 | 51.80 ±1.82 | 39.00 ±6.37 |
| CGMatch (CVPR'25) | 40.01 ±18.31 | 31.78 ±16.07 | 59.74 ±14.41 | 11.02 ±0.80 | 10.24 ±1.16 | 10.27 ±0.53 | 60.81 ±3.15 | 31.98 ±7.78 |
| Meta-Expert (ICML'25) | 72.38 ±4.35 | 45.74 ±3.15 | 50.47 ±24.94 | 6.61 ±0.30 | 5.61 ±0.24 | 6.66 ±0.07 | 34.09 ±2.51 | 31.65 ±5.08 |
| CPG (NeurIPS'25) | 54.04 ±24.50 | 42.90 ±14.46 | 57.39 ±6.54 | 11.65 ±1.02 | 11.32 ±0.16 | 12.15 ±0.42 | 48.98 ±1.72 | 34.06 ±6.98 |
| SAGE (Ours) | **93.39** ±2.27 | **86.93** ±2.48 | **84.11** ±2.81 | **13.59** ±1.05 | **12.88** ±0.58 | **12.88** ±0.39 | **64.90** ±1.82 | **52.67** ±1.63 |

triggered by extreme label scarcity. SAGE effectively mitigates this by anchoring the representation space with a rigid coordinate frame, which guides inter-class representation separation and prevents representation confusion even when initial supervision is extremely scarce.

Overall, SAGE establishes new state-of-the-art benchmarks across all evaluated scenarios. Compared to previous leading methods, SAGE achieves an average accuracy improvement of **8.52** pp across five benchmark datasets.

### 4.4. Ablation Study

To verify the effectiveness of each component in our SAGE, we conduct comprehensive ablation studies on CIFAR-10 and CIFAR-100, as shown in Table 4.

**Architectural isolation safeguards the primary decision boundary.** Introducing the auxiliary branch (AB) for decoupling improves the average performance from 45.56% to 49.58% (+4.02 pp). This gain confirms that isolating the learning process of unlabeled data helps shield the primary classification head from unreliable gradients during training, thereby preserving the purity of its task-specific decision boundary and improving the pseudo-label accuracy (+13.35 pp from V1 → V2, as shown in Fig. 4).

**Distribution-agnostic prioritization filters noise without distribution priors.** The integration of distribution-agnostic reliability prioritization (DRP) yields a further substantial gain of +5.43 pp, bringing the average accuracy to 55.01%. By evaluating both absolute certainty and relative discrim-

*Table 4.* Comparison of accuracy (%) with and without key components of the proposed method under $\gamma_l = \gamma_u$ (i.e., unlabeled data distribution is uniform) and $\gamma_l \neq \gamma_u$ (i.e., unlabeled data distribution is arbitrary) settings.

| Ablations | | | CIFAR-10 | | | | CIFAR-100 | | | | Average Accuracy |
|---|---|---|---|---|---|---|---|---|---|---|---|
| | | | $N=40, M=13980$ | | $N=40, M=12390$ | | $N=400, M=24600$ | | $N=400, M=19400$ | | |
| w/ AB | w/ DRP | w/ GRI | $\gamma_u=1$ | $\gamma_u=50$ | $\gamma_u=1$ | $\gamma_u=100$ | $\gamma_u=1$ | $\gamma_u=5$ | $\gamma_u=1$ | $\gamma_u=10$ | |
| | | | 64.22 | 57.24 | 67.16 | 50.75 | 33.54 | 33.30 | 29.16 | 29.12 | 45.56 |
| ✓ | | | 79.20 | 64.52 | 73.85 | 54.38 | 33.78 | 32.54 | 29.32 | 29.01 | 49.58 ↑ 4.02 |
| ✓ | ✓ | | 81.30 | 68.99 | 81.64 | 64.26 | 38.13 | 37.00 | 34.62 | 34.14 | 55.01 ↑ 9.45 |
| ✓ | ✓ | ✓ | **82.68** | **69.64** | **81.90** | **66.83** | **43.45** | **40.77** | **40.72** | **37.15** | **57.89** ↑ 12.33 |

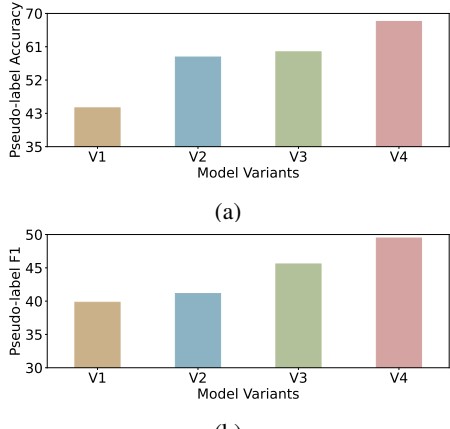

(a)

(b)

*Figure 4.* Ablation study demonstrating the incremental gains in pseudo-label accuracy (a) and F1 score (b) achieved by our modules under an arbitrary unlabeled data distribution: V1 (baseline), V2 (+ auxiliary branch), V3 (+ distribution-agnostic reliability prioritization), and V4 (+ graph-state relational inference).

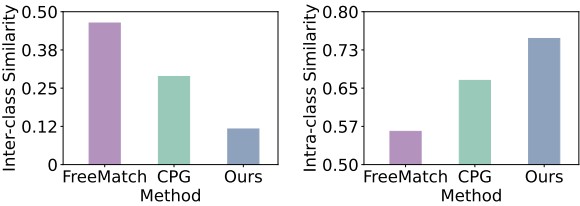

*Figure 5.* Quantitative evaluation of representation quality via inter-class similarity (↓) and intra-class similarity (↑). The dataset is SVHN with $(N_{max}, M_{max}, \gamma_l, \gamma_u) = (4, 4996, 1, 150)$. The results demonstrate that SAGE achieves the lowest inter-class overlap and the highest intra-class compactness, validating the effectiveness of fixed simplex equiangular anchors as a stable coordinate frame.

inability, DRP effectively prioritizes reliable pseudo-labels. This distribution-agnostic weighting is critical in UniSSL, as it filters erroneous signals without relying on inaccessible distribution priors.

**Structural consensus bypasses distribution estimation through high-order dependencies.** The full version of our SAGE, incorporating the graph-state relational inference (GRI) module, achieves a peak accuracy of 57.89% (a total improvement of +12.33 pp over the baseline). By mining high-order inter-sample dependencies across the data manifold, GRI establishes a stable structural consensus. This mechanism not only bypasses distribution estimation but also establishes a virtuous cycle where robust structural guidance enhances pseudo-label quality (+7.98 pp and +3.9 pp for V3 → V4 in accuracy and F1 score, respectively, as shown in Fig. 4). Further analysis of the effects of loss terms within the GRI module is detailed in Appendix E.

The results demonstrate that each component is effective individually, and their integration drives higher performance gains.

### 4.5. Further Analysis

**Geometric anchors promote representation discriminability.** To quantitatively evaluate how the fixed coordinate frame influences the representation space, we analyze the inter-class and intra-class similarities in Fig. 5. Existing methods like FreeMatch and CPG exhibit high inter-class similarity and low intra-class similarity, suggesting significant representation confusion. In contrast, our SAGE achieves the lowest inter-class similarity and the highest intra-class similarity. This superior performance stems from anchoring representations to a fixed coordinate frame and forcing the backbone to guide different class representations toward maximally separated directions.

**Robustness under extreme label scarcity and imbalance.** We evaluate SAGE on CIFAR-10 under an extreme UniSSL setting with $(N = 10, M = 12390, \gamma_l = 1, \gamma_u = 100)$, i.e., only one labeled sample per class. SAGE achieves 43.02%, outperforming CGMatch (37.31%), Meta-Expert (17.84%), and CPG (32.12%). In particular, it surpasses the strongest prior baseline by 5.71 pp, demonstrating that relation-based structural inference remains effective even under severe label scarcity. We also analyze the failure boundary under extreme imbalance. Since the anchors in SAGE are fixed frequency-agnostic geometric references rather than learned class prototypes, they do not inherently favor majority classes. Combined with the discriminative representations in Fig. 2(g) under an imbalance ratio of 150, these results suggest that SAGE remains robust in highly imbalanced settings. Potential failure cases may instead arise when extremely sparse minority class samples, severe manifold overlap, or overly strong graph diffusion cause structural consensus to be dominated by majority class neighborhoods.

**Good generalization to large-scale datasets.** We evaluate SAGE on the large-scale dataset ImageNet-127 (Deng et al., 2009) using WRN-28-2 under a semi-supervised setting, where $\gamma_l = \gamma_u = 286$, 10% of the training data is used as labeled data and the remaining 90% as unlabeled data. SAGE achieves an accuracy of 53.83%, outperforming ACR (47.61%), SimPro (46.93%), CDMAD (21.92%), Meta-Expert (49.83%), and CPG (50.43%). In particular, it surpasses the strongest prior baseline CPG by 3.40 pp. These results suggest that SAGE generalizes well to large-scale datasets.

**Robustness under open-set settings.** To further evaluate the robustness of SAGE, we conduct an open-set experiment on CIFAR-10 with ($N = 40, M = 12390, \gamma_l = 1, \gamma_u = 100$), where samples from 10 unknown ImageNet-127 classes are additionally introduced into the unlabeled dataset. SAGE achieves 60.88% accuracy, consistently outperforming SSB (Fan et al., 2023) (47.41%), SCO-Match (Wang et al., 2024) (51.03%), and CaliMatch (Bae et al., 2025) (56.12%). In particular, SAGE surpasses the strongest baseline CaliMatch by 4.76 pp. These results indicate that SAGE remains robust under open-set scenarios, benefiting from representation learning on unlabeled data while reducing the influence of noisy pseudo-labels from unknown classes.

**Robust to representation and anchor variations.** To examine robustness to representation dimensionality and the number of anchors, we conduct experiments on CIFAR-10 under the setting ($N = 40, M = 12390, \gamma_l = 1, \gamma_u = 100$). For the former, we replace the backbone with ResNet-50 and use a 2048-dimensional output representation. Under this setting, SAGE achieves 35.93%, outperforming CGMatch (22.08%), Meta-Expert (31.85%), and CPG (24.74%), and surpassing the strongest prior baseline by 4.08 pp. This demonstrates that the method remains effective under substantially higher-dimensional representations. Moreover, under this extremely label-scarce setting, the early representations learned by larger models are typically not yet well separated. The results show that our relation-based structural inference remains effective even when the representations are less well separated, suggesting that SAGE generalizes well to larger architectures. We further study the sensitivity to the number of anchors $K$. When varying $K$, the model achieves 65.30% ($K = 50$), 63.98% ($K = 80$), 66.83% ($K = 129$), and 63.87% ($K = 200$). Despite this wide range of configurations, performance varies only moderately and remains consistently high, indicating robustness to anchor design.

**Statistical analysis validates the superiority of SAGE over diverse baselines.** We summarize the win/tie/loss counts between our SAGE and nine baseline methods across different datasets under varying unlabeled data distributions,

using a pairwise t-test at a 0.05 significance level. The results show that our SAGE outperforms baseline methods in all cases, with a significant improvement in approximately 70% of cases (136 out of 195), demonstrating its effectiveness. The win/tie/loss counts are detailed in Appendix F.

Computational overhead analysis is detailed in Appendix G. Scalability and broader applicability analysis is detailed in Appendix H.

## 5. Conclusion

In this paper, we formalized Universal Semi-supervised Learning (UniSSL) to address the challenges of unknown, arbitrary unlabeled data distributions under extreme label scarcity. We proposed Simplex Anchored Graph-state Equipartition (SAGE), a framework that shifts the paradigm from distribution estimation to representation-level structural inference. By leveraging fixed simplex equiangular anchors, SAGE establishes a stable coordinate frame that grounds representation learning and guides inter-class separation. Through the integration of Graph-state Relational Inference for structural consensus, Distribution-agnostic Reliability Prioritization for adaptive sample weighting, and an auxiliary branch for architectural isolation, our approach overcomes representation confusion. Extensive evaluations across five benchmarks demonstrate that SAGE achieves state-of-the-art performance, yielding an average improvement of 8.52% over competing baselines and providing a practical framework for realistic scenarios. Despite its strengths, SAGE assumes a fixed number of known classes and its structural affinity matrix scales quadratically with batch size. Future work may explore adaptive anchor generation and more efficient graph diffusion for open-world and large-scale scenarios.

## Acknowledgements

This work was supported by the National Natural Science Foundation of China under Grants U24A20322 and 62576094. This research work is also supported by the Big Data Computing Center of Southeast University.

## Impact Statement

This paper presents work whose goal is to advance the field of Machine Learning. Specifically, we propose SAGE, a framework that shifts the paradigm from distribution estimation to representation-level structural inference, effectively overcoming representation confusion under extreme label scarcity and arbitrary distributions. There are many potential societal consequences of our work, none which we feel must be specifically highlighted here.

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

## Table of Contents for the Appendix

# A. Additional Related Work

**Neural collapse** (NC) reveals that during the terminal phase of training, the last-layer features of the same class tend to collapse into their within-class means, eventually forming a simplex equiangular tight frame (ETF) (Papyan et al., 2020). The simplex ETF exhibits equiangular separation, providing an optimal geometric structure for discriminative classification. Recently, the NC phenomenon has been investigated in various scenarios such as long-tailed learning (Li et al., 2022; Zhang et al., 2024), out-of-distribution detection (Xiao et al., 2024; Liu & Qin, 2025), and noisy label learning (Nguyen et al., 2023). In this work, we leverage the simplex ETF as a fixed coordinate frame to anchor the representation space. By anchoring representations to these fixed geometric anchors, we guide inter-class representation separation and establish a foundation for structural inference.

# B. Pseudo-code of the Proposed SAGE

We present the pseudo-code of our SAGE in Algorithm 1.

---

**Algorithm 1** Training Process of the Proposed Method

---

1: **Input**: Labeled dataset $\mathcal{D}_l$, unlabeled dataset $\mathcal{D}_u$, regularization parameter $\lambda$, and graph transition steps $\beta$.
2: **Output**: Optimized model parameters $\theta$.
3: Generate the fixed simplex equiangular anchors $\mathbf{P}$ via Eq. (5);
4: Initialize model parameters $\theta$ randomly;
5: **for** iteration $t = 1, 2, \ldots, T$ **do**
6:     Calculate supervised loss $\mathcal{L}_{cls}$ for primary classifier $\phi_{cls}$ via Eq. (7);
7:     **// Graph-state Relational Inference (GRI)**
8:     Compute relational embeddings $\mathbf{a}_i$ for unlabeled samples via Eq. (2);
9:     Construct row-normalized state transition matrix $\hat{\mathbf{P}}$ by calculating structural affinity matrix $\mathbf{A}$ using $\mathbf{a}_i$;
10:     Derive structural consensus $\mathbf{G} = \hat{\mathbf{P}}^\beta$;
11:     Calculate structural contrastive loss $\mathcal{L}_{con}$ using $\mathbf{G}$ via Eq. (3);
12:     Calculate representation consistency loss $\mathcal{L}_{sim}$ via Eq. (4);
13:     **// Distribution-agnostic Reliability Prioritization (DRP)**
14:     Update independent EMA statistics $(\mu_\kappa, \sigma_\kappa^2)$ for $\kappa \in \{max, gap\}$;
15:     Compute adaptive weight $w = \mathcal{W}(q_{max}; \mu_{max}, \sigma_{max}) \cdot \mathcal{W}(q_{gap}; \mu_{gap}, \sigma_{gap})$ via Eq. (6);
16:     Calculate auxiliary loss $\mathcal{L}_{aux}$ for auxiliary classifier $\phi_{aux}$ via Eq. (8);
17:     **// Optimization**
18:     Obtain total loss $\mathcal{L}_{total}$ via Eq. (9);
19:     Update model parameters $\theta$ via gradient descent;
20: **end for**

---

# C. Dataset Details

We conduct our experiments on five widely-used datasets (i.e., CIFAR-10 (Krizhevsky, 2009), CIFAR-100 (Krizhevsky, 2009), Food-101 (Bossard et al., 2014), SVHN (Netzer et al., 2011), and STL-10 (Coates et al., 2011)), following the main experimental settings in FreeMatch (Wang et al., 2023) and CPG (Hou et al., 2025), with details provided below.

**CIFAR-10**: We evaluate nine experimental settings with $(N_{max}, M_{max}) = (4, 4996)$. While the labeled data follows a uniform distribution (i.e., $\gamma_l = 1$), the unlabeled data is drawn from uniform, long-tailed, or arbitrary distributions. For the imbalanced scenarios (i.e., long-tailed and arbitrary), we vary the imbalance ratio $\gamma_u \in \{50, 100, 150\}$.

**CIFAR-100**: Similarly, for CIFAR-100, we adopt an identical distribution pattern but adjust the dataset split to $(N_{max}, M_{max}) = (4, 496)$ and set the imbalance ratio $\gamma_u \in \{5, 10, 15\}$.

**Food-101**: We use the same dataset split as CIFAR-100 with the imbalance ratio $\gamma_u = 15$.

**SVHN**: We adopt the setting of $(N_{max}, M_{max}) = (4, 4996)$, with a specific imbalance ratio $\gamma_u = 150$.

**STL-10**: Since the ground-truth labels of the unlabeled data in STL-10 are unknown, we only set $(N_{max}, \gamma_l) = (4, 1)$ and directly utilize the original unlabeled set.

# D. Parameter Sensitivity Analysis

We evaluate the sensitivity of two key hyperparameters (i.e., $\lambda$ and $\beta$) on CIFAR-10 ($N = 40, M = 12390, \gamma_l = 1, \gamma_u = 100$) under an arbitrary unlabeled data distribution to demonstrate the robustness of our SAGE.

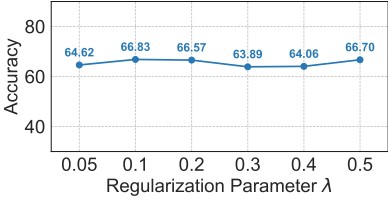 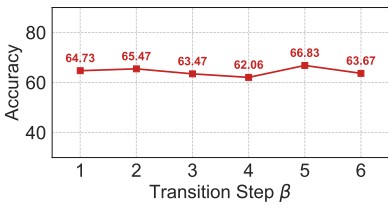

*Figure 6.* Parameter sensitivity analysis of our SAGE on CIFAR-10 under an arbitrary unlabeled data distribution. The results demonstrate that SAGE is robust to the choice of $\lambda$ due to the fixed coordinate frame, and that $\beta = 5$ provides the optimal diffusion of structural consensus to the representation space without over-smoothing.

**Regularization term** $\lambda$. The parameter $\lambda$ controls the density of the relational embedding and prevents its degeneration into sparse assignments. As shown in Fig. 6 (left), SAGE achieves a peak performance of 66.83% at $\lambda = 0.1$, indicating that this magnitude provides an appropriate density for capturing high-order inter-sample dependencies. Although the accuracy exhibits fluctuations as $\lambda$ increases up to 0.5 (e.g., dropping to 63.89% at $\lambda = 0.3$ before recovering to 66.70% at $\lambda = 0.5$), the overall performance remains consistently superior to the baselines. This reinforces that the fixed simplex equiangular anchors establish a stable coordinate frame, ensuring the model's robustness across a wide range of regularization strengths.

**Graph transition steps** $\beta$. The parameter $\beta$ determines the number of graph transition steps used to derive the structural consensus. As illustrated in Fig. 6 (right), $\beta = 5$ is the optimal setting. Lower values (i.e., $\beta \leq 4$) yield suboptimal results as they limit the diffusion of relational information to local neighborhoods, which is insufficient to capture high-order inter-sample dependencies across the data manifold. Conversely, increasing $\beta$ beyond 5 leads to a performance drop due to the over-smoothing effect, where the propagated structural consensus begins to blur inter-class boundaries. Thus, a 5-step transition strikes the optimal balance between structural consensus propagation and semantic discriminability.

# E. Analysis of the Effects of Loss Terms within the GRI Module

We evaluate the synergy between the structural contrastive loss ($\mathcal{L}_{con}$) and the representation consistency loss ($\mathcal{L}_{sim}$) within the GRI module. As illustrated in Fig. 7, the full GRI configuration achieves peak performance across all evaluated scenarios (e.g., 69.64% at $\gamma_u = 50$). Notably, the exclusion of $\mathcal{L}_{con}$ results in a more pronounced performance degradation (dropping to 67.78%) compared to the removal of $\mathcal{L}_{sim}$ (dropping to 68.22%). These findings indicate that while both terms are complementary, $\mathcal{L}_{con}$ serves as the primary mechanism for leveraging structural consensus, providing stable structural guidance. Conversely, $\mathcal{L}_{sim}$ acts as a vital regularizer that ensures the local smoothness of the representation space, consistent with our objective to stabilize the data manifold across diverse augmentation views.

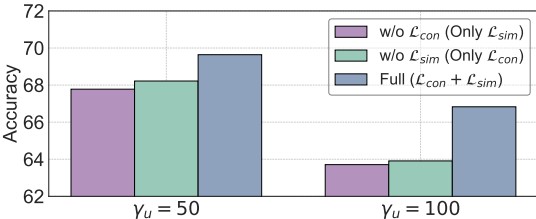

*Figure 7.* Ablation study of the loss terms within the GRI module. The results under arbitrary unlabeled data distributions across different imbalance ratios demonstrate the synergistic effect between structural contrastive loss ($\mathcal{L}_{con}$) and representation consistency loss ($\mathcal{L}_{sim}$), where $\mathcal{L}_{con}$ plays a more pivotal role in providing stable structural guidance.

# F. Statistical Significance

Table 5 presents the win/tie/loss counts between our SAGE and nine baseline methods across different datasets under varying unlabeled data distributions, using a pairwise t-test at a 0.05 significance level. The results show that our SAGE outperforms baseline methods in all cases, with a significant improvement in approximately 70% of cases (136 out of 195), demonstrating its effectiveness.

*Table 5.* Statistical significance of performance differences assessed with a pairwise t-test at a 0.05 significance level, reported as win/tie/loss counts.

| Method | Uniform | Long-tailed | Arbitrary | Total |
|---|---|---|---|---|
| FixMatch (NeurIPS'20) | 4 / 5 / 0 | 5 / 3 / 0 | 2 / 6 / 0 | 11 / 14 / 0 |
| FreeMatch (ICLR'23) | 5 / 2 / 0 | 5 / 1 / 0 | 5 / 1 / 0 | 15 / 4 / 0 |
| SoftMatch (ICLR'23) | 5 / 2 / 0 | 4 / 2 / 0 | 6 / 0 / 0 | 15 / 4 / 0 |
| CGMatch (CVPR'25) | 3 / 6 / 0 | 6 / 2 / 0 | 3 / 5 / 0 | 12 / 13 / 0 |
| ACR (CVPR'23) | 5 / 2 / 0 | 4 / 2 / 0 | 3 / 3 / 0 | 12 / 7 / 0 |
| SimPro (ICML'24) | 7 / 0 / 0 | 6 / 0 / 0 | 6 / 0 / 0 | 19 / 0 / 0 |
| CDMAD (CVPR'24) | 6 / 1 / 0 | 5 / 1 / 0 | 4 / 2 / 0 | 15 / 4 / 0 |
| Meta-Expert (ICML'25) | 9 / 0 / 0 | 8 / 0 / 0 | 5 / 3 / 0 | 22 / 3 / 0 |
| CPG (NeurIPS'25) | 8 / 1 / 0 | 6 / 2 / 0 | 1 / 7 / 0 | 15 / 10 / 0 |
| Total | 52 / 19 / 0 | 49 / 13 / 0 | 35 / 27 / 0 | 136 / 59 / 0 |

## G. Computational Overhead

Our method introduces only limited additional computational cost and remains comparable to recent baselines in terms of training time. On the same RTX 4090 GPU, the total training time is 18.5 h for FixMatch, 29.5 h for CGMatch, 19.2 h for Meta-Expert, 22.6 h for CPG, and 18.8 h for our method.

The additional overhead is small for two main reasons. First, the anchor set is very small, with the number of anchors set to the output representation dimension plus one. In our default setting, we use only 129 anchors for a 128-dimensional representation. Therefore, the anchor-related computation is lightweight by design. Moreover, these anchors are fixed after construction rather than learned during training. They are generated only once before training, taking just 0.5 s (0.0007% of the total training time), and introduce no iterative updates or additional optimization cost. Second, the additional computation during training is also lightweight. The main extra cost comes from the batch-level relational inference in GRI, which is performed within each mini-batch rather than over the full unlabeled dataset. In our setting, this reduces to a single matrix multiplication between a $448 \times 128$ matrix and a $128 \times 129$ matrix. The total runtime of the GRI module is only 0.007 s per iteration. Since this operation involves only mini-batch features and a small fixed anchor set, its practical overhead is very limited.

Overall, both anchor construction and relational inference are efficient, making the runtime of our method comparable to that of recent baselines. Specifically, when the output representation dimension is 128 and the batch size is 448, the time required for anchor construction and relational inference is consistently about 0.5 s and 0.007 s, respectively, regardless of the dataset.

## H. Scalability and Broader Applicability

UniSSL remains relevant in the era of large language models (LLMs), as the fundamental challenge persists: labeled data are scarce, while unlabeled data are abundant. Recent studies on LLM adaptation also highlight semi-supervised fine-tuning as an effective data-efficient learning paradigm (Yang et al., 2026).

In this context, UniSSL is not designed to compete with LLMs, but to provide a general principle for data-efficient learning under scarce supervision and potentially unknown unlabeled data distributions. By shifting supervision from unreliable pseudo-labels to inter-sample structural consensus, our method is not tied to a specific backbone and may generalize to broader settings with diverse or distribution-mismatched unlabeled data.

This perspective is aligned with recent advances in LLM and agent learning, where practical bottlenecks arise not only from model capacity, but also from adaptation under limited reliable supervision and abundant noisy or weakly labeled data. Such settings are common in LLM post-training and domain adaptation, agent learning from interaction trajectories, and open-world deployment. For example, semi-supervised fine-tuning for LLMs (Luo et al., 2025) and noisy-feedback filtering in RLHF (Zhang et al., 2025) both emphasize learning under imperfect supervision signals, while web agents (He et al., 2024) and retrieval-based domain adaptation methods (Long et al., 2023) further demonstrate the importance of robust learning under distribution shifts.

We therefore view UniSSL as complementary to these LLM and agent systems. As these systems are deployed in broader and noisier environments, semi-supervised learning remains a general and effective paradigm for learning from scarce annotations and abundant imperfect data.

