# OpenReview forum: "Beyond Distribution Estimation: Simplex Anchored Structural Inference Towards Universal Semi-Supervised Learning"
_ICML.cc/2026/Conference — ICML 2026 regular_

### Official Review · Reviewer_Lapm · 2026-03-08

**Soundness:** 3
**Presentation:** 3
**Significance:** 2
**Originality:** 2
**Overall Recommendation:** 4
**Confidence:** 4

**Summary:**

This paper studies a universal semi-supervised learning setting where the label and unlabel ratio is varied across classes. To tackle this problem through pseudo-labeling, this paper proposes three main technical contributions: 1. The Graph-state Relational Inference (GRI), which treats unlabeled data as nodes in a graph, by inspecting high-order dependencies in the geometrical aspect, the model can effectively correct noisy labels. 2. A noisy label filtering process, which leverages both confidence and prediction margin to remove inconfident or uncertain predictions. 3. Moreover, the proposed method separates the clean data classification from noisy data identification to ensure that the final decision is not contaminated by the noisy unlabeled data. By combining all three techniques, the proposed method has shown effective learning performance that surpasses many existing SOTA methods.

**Compliance With Llm Reviewing Policy:**

Affirmed.

**Final Justification:**

The authors have addressed most of my questions with careful explanation and experiments. Therefore I am willing to raise my score.

**Key Questions For Authors:**

- Is there practical guidance on how to set the hyperparameters?
- Can the proposed method handle out-of-distribution data? Does the UniSSL setting conflict with existing works?
- What is the key motivation of exploiting graph information?
- How can the proposed framework be scaled to wider applications? What is the purpose of the proposed UniSSL under the era of LLMs?

**Limitations:**

There are many limitations; however, this paper did not address any. It demonstrated superior performance under the UniSSL setting, but whether it remains effective under a standard SSL setting remains uncertain. If not, the problem can be easily solved by importance reweighting to recover the normal SSL distribution.

For more limitations, please consider the ones raised in weaknesses and questions.

**Strengths And Weaknesses:**

Strengths:
- This paper has many technical contributions, which leverage many state-of-the-art techniques such as diffusion and geometric learning, further benefiting the high-level understanding of the data, thus improving the identification of the noisy labels.
- The experimental improvements are quite promising, and the evaluation spectrum widely covers many areas.
- The research problem is interesting and has practical significance.

Weaknesses:
- Although the proposed method can be conducted in a single GTX4090, however, the methodology design appears to be quite combinational which lacks compactivity. More importantly, the deployment of diffussion process, structural affinity matrix, and graph transition makes it seriously difficult to scale.
- There are many hyperparameters are introduced, such as egularization parameter $\lambda$, the transition steps $\beta$, or the EMA decay rates for DRP. Given the complexity of the proposed UniSSL setting, it would be seriously hard to settle an optimal hyperparameter setting.
- The proposed UniSSL setting seems to be already existing in Huang et al., Universal Semi-Supervised Learning, NeurIPS 2021, which considers open-set data with varied label distribution. However, this paper did not distinguish their difference, meanwhile, whether the proposed method can handle open-set SSL is unknown. Therefore, the name UniSSL seems to be a little overclaiming.
- Moreover, Graph-based SSL has also been discussed with a long history, Song et al., Graph-based Semi-supervised Learning. What is the foundamental intuition why the proposed method that exploits graph information can outperform existing graph-based SSLs? In my understanding, it seems that due to the employed SOTA techniques, such as diffusion, further enables performance enhancement. In this case, the methodological innovation would be limited. It requires further justification of the core contribution to be further assessed.

---

> ### Author Rebuttal · Authors · 2026-03-30
>
> We appreciate the reviewer’s insightful comments and respond to each point below.
>
> ## Method Complexity and Scalability (W1)
>
> Our method is **not a naive combination of existing modules**, but is **designed for a key failure mode in UniSSL**: when labeled data are extremely scarce and the unlabeled data distribution is unknown, pseudo-labels become unreliable. Prior works [1,2] show that self-supervised learning yields robust representations. Based on this, we argue that **inter-sample relations are more reliable than pseudo-labels** in UniSSL, and thus shift supervision from pseudo-labels to **representation-level structural consensus**.
>
> **The method is also efficient.** On the same RTX 4090, our total training time is **18.8 h**, compared with **18.5 h** for the much simpler FixMatch. Anchor construction is done once before training, taking **0.5 s** (**0.0007%** of total training time). The main extra cost during training is a batch-wise matrix multiplication (**448×128 by 128×129**), which takes only **0.007 s** per iteration. Thus, the practical overhead is **negligible**. Please see our response to **Reviewer dZFX’s W3** for detailed runtime analysis.
>
> [1] NeurIPS’22-Contrastive and Non-Contrastive Self-Supervised Learning Recover Global and Local Spectral Embedding Methods
>
> [2] CVPR’21-Understanding the Behaviour of Contrastive Loss
>
> ## Hyperparameter Sensitivity (W2&Q1)
>
> We agree that hyperparameter sensitivity is important in UniSSL. However, our hyperparameters are **fixed across datasets and settings**, rather than tuned case by case. Specifically, we set the EMA decay in DRP to **0.999**, following common SSL practice. We also provide a sensitivity analysis of **$\lambda$** and **$\beta$** in **Appendix D / Fig. 6**, showing that SAGE is **reasonably robust**.
>
> ## UniSSL Terminology and Open-Set Setting (W3&Q2)
>
> Our **UniSSL** setting differs from Huang et al. (NeurIPS’21). Their setting focuses on **open-set SSL with uniform distributions**, where unlabeled data include samples from unknown classes, whereas ours addresses a **broader range of unknown unlabeled data distributions**. Although the original submission did not explicitly evaluate the open-set case, our method can also handle it.
>
> To verify this, we added an **open-set experiment** and compared with recent baselines. On CIFAR-10 with **$(N=40,M=12390,\gamma_l=1,\gamma_u=100)$**, we further added samples from **10 unknown ImageNet classes** to the unlabeled dataset. As shown in **Table R9**, our method remains effective, outperforming **SSB**, **SCOMatch**, and **CaliMatch** by **13.47**, **9.85**, and **4.76** points, respectively. This is because our method uses unlabeled data mainly to learn robust representations rather than relying heavily on pseudo-labels, while the main branch is trained only on labeled data and is thus less affected by noisy pseudo-labels from unknown classes. We will revise the paper to clarify the difference from prior Universal SSL / open-set SSL settings.
>
> **Table R9 Open-set**
> |SSB (ICCV’23)|SCOMatch (ECCV’24)|CaliMatch (ICCV’25)|SAGE|
> |---|---|---|---|
> |47.41|51.03|56.12|**60.88**|
>
> ## Motivation for Exploiting Graph Information (W4&Q3)
>
> Classical graph-based SSL mainly uses graphs for **graph construction and label propagation**, enforcing label smoothness under the manifold assumption. Our motivation is different: in **UniSSL**, **pseudo-labels become unreliable** when labels are extremely scarce and unlabeled data distributions are unknown. We therefore use graph information not for conventional label propagation, but to capture **representation-level structural consensus**, which is more reliable than pseudo-label predictions in this regime. Thus, our contribution is **not diffusion itself**, but shifting supervision from **pseudo-labels** to **inter-sample relations**.
>
> ## Scalability and Broader Applicability (Q4)
>
> UniSSL remains relevant in the **LLM era** because the core challenge persists: **labeled data are scarce, while unlabeled data are abundant**. A recent LLM study [1] also highlights **semi-supervised fine-tuning** as a data-efficient adaptation strategy.
>
> Thus, UniSSL is **not meant to compete with LLMs**, but to provide a general principle for **data-efficient learning under scarce labels and unknown unlabeled data distributions**. By shifting supervision from unreliable pseudo-labels to **inter-sample structural consensus**, our method is not tied to a specific backbone and may generalize to broader settings with diverse or **distribution-mismatched** unlabeled data.
>
> [1] ICLR’26-TraPO: A Semi-Supervised Reinforcement Learning Framework for Boosting LLM Reasoning
>
> ## Performance under Standard SSL (L1)
>
> Our method is also effective under **standard SSL**. In our paper, the **uniform** setting corresponds to standard SSL, where both labeled and unlabeled data follow uniform distributions (**$\gamma_l=\gamma_u=1$**). Under this setting, SAGE still achieves strong performance.

---

> > ### Author Rebuttal · Reviewer_Lapm · 2026-04-03
> >
> > Thanks for the detailed rebuttal from the authors. The rebuttal justified that the method is efficient, robust, and motivated by representation-level supervision, with evidence for open-set and standard SSL performance. After careful consideration, most of my concerns are addressed. Therefore, I am willing to raise my score.
> >
> > However, I have one follow-up question: I hope the authors could further justify the significance of semi-supervised learning in the era of LLMs and Agents.

---

> > > ### Author Response · Authors · 2026-04-04
> > >
> > > We thank the reviewer for the positive update and are glad that our rebuttal helped clarify the concerns. We sincerely thank the reviewer for the thoughtful follow-up and valuable suggestions, which will help improve the paper.
> > >
> > > Regarding this important follow-up, we agree that the broader significance should be clarified more explicitly. In the era of LLMs and agents, the practical bottleneck is often not model capacity alone, but how to adapt models with limited reliable supervision and abundant unlabeled or weakly labeled data, which is often noisy and distribution-mismatched. This regime naturally appears in LLM post-training and domain adaptation, agent learning from interaction trajectories, and open-world deployment.
> > >
> > > Our method is designed exactly for this setting. Rather than relying purely on pseudo-label predictions, it exploits representation-level structural consensus and inter-sample relations, which are more robust when pseudo-labels are unreliable and unlabeled data may contain samples from unknown classes. Therefore, we view our contribution as complementary to LLM and agent systems: as these systems are deployed in broader and noisier environments, semi-supervised learning remains important as a data-efficient paradigm for learning from scarce annotations and abundant imperfect data. We will clarify this broader significance more explicitly in the revision.
> > >
> > > We will also add discussion and references to recent work on semi-supervised LLM fine-tuning, noisy-feedback filtering in RLHF, web agents, and retrieval-based domain adaptation in the revision, including:
> > >
> > > [1] TraPO: A Semi-Supervised Reinforcement Learning Framework for Boosting LLM Reasoning, ICLR 2026.
> > >
> > > [2] Semi-supervised Fine-tuning for Large Language Models, NAACL 2025.
> > >
> > > [3] Policy Filtration for RLHF to Mitigate Noise in Reward Models, ICML 2025.
> > >
> > > [4] WebVoyager: Building an End-to-End Web Agent with Large Multimodal Models, ACL 2024.
> > >
> > > [5] Adapt in Contexts: Retrieval-Augmented Domain Adaptation via In-Context Learning, EMNLP 2023.

---

### Official Review · Reviewer_Af3X · 2026-03-09

**Soundness:** 3
**Presentation:** 2
**Significance:** 3
**Originality:** 3
**Overall Recommendation:** 4
**Confidence:** 3

**Summary:**

This paper addresses the challenge of Universal Semi-supervised Learning (UniSSL), characterized by extremely scarce labeled data and unlabeled data following unknown, arbitrary distributions (e.g., long-tailed, balanced, or random). It identifies the issue of "representation confusion" in traditional methods caused by their over-reliance on distribution assumptions and, accordingly, proposes the SAGE (Simplex Anchored Graph-state Equipartition) framework.

This method innovatively leverages structural consensus among samples to correct erroneous pseudo-labels and introduces Simplex Anchors to enforce class separation within the feature space. Combined with a Distribution-agnostic Reliability Prioritization (DRP) mechanism and an auxiliary branch design, SAGE effectively enhances model robustness in extreme environments.

**Compliance With Llm Reviewing Policy:**

Affirmed.

**Final Justification:**

The authors have addressed most of my concerns, so I will keep my positive score.

**Key Questions For Authors:**

1 The paper sets $K$ anchors to be $(d+1)$-dimensional. If the number of classes $C$ in a classification task is much smaller or larger than the representation dimension $d$, how does this fixed geometric structure adaptively guide the feature separation for different numbers of categories?

2The paper observes that inter-sample relations are more reliable than pseudo-labels. Is there a critical tipping point where, if labels become scarce to a certain extent (e.g., only 1 label per class with an extremely skewed distribution), this relation-based inference would also completely fail?

3The number of comparison algorithms selected in Table 3 is fewer than those in Tables 1 and 2; an explanation is required to clarify the reasons for this discrepancy.

4 The observation mentioned in the third paragraph of the Introduction—stating that "while pseudo-labels frequently fail, the inter-sample relations captured in the representation space remain robust and preserve consistent semantics"—lacks support from the preceding text. This conclusion appears somewhat abrupt, and corresponding explanations need to be added.

**Limitations:**

The Conclusion section does not summarize the limitations of this paper and future research directions.

**Strengths And Weaknesses:**

Strengths:

The paper is well-structured, and the proposed Simplex Anchored Graph-state Equipartition (SAGE) method is technically sound. By performing representation-level structural inference to bypass distribution estimation and employing simplex equiangular tight frame vectors to guide inter-class separation, it effectively mitigates the representation confusion issue caused by existing pseudo-labeling methods’ reliance on distribution assumptions. Furthermore, the work shifts the research focus from pseudo-labels to representation-level structural inference, this shift provides a novel perspective for addressing the Universal Semi-supervised Learning (UniSSL) problem.

Weaknesses

1 Although SAGE is designed to rectify pseudo-labeling errors, its core mechanism relies heavily on the "inter-sample relations" within the representation space. If the feature representations are extremely poor in the early stages of training, the resulting structural consensus may also be biased, potentially leading the model into an incorrect optimization direction.

2 The generation of Simplex Anchors involves QR decomposition and eigenvalue decomposition. While current experiments are conducted on image classification datasets (such as CIFAR and SVHN), there may be scalability limitations when calculating these closed-form solutions or performing large-scale matrix multiplications in ultra-high-dimensional representation spaces.

---

> ### Author Rebuttal · Authors · 2026-03-30
>
> We appreciate the reviewer’s insightful comments and respond to each point below.
>
> ## Robustness of Relations under Poor Early Representations (W1)
>
> Our key motivation is that inter-sample relations provide a **more stable supervisory signal than pseudo-labels** under UniSSL. To address this concern, we further evaluate SAGE with **ResNet-50 on CIFAR-10 with $(N=40,M=12390,\gamma_l=1,\gamma_u=100)$**, which serves as a **more challenging stress test** under the same extremely label-scarce setting, as the model needs to learn discriminative representations from very limited supervision (**1 labeled sample per class**). As shown in **Table R5**, SAGE still outperforms the strongest prior baseline, which mainly relies on pseudo-labels, by **4.08 points**. This result further supports that our relation-based structural inference remains effective even when the early representations are less well separated.
>
> **Table R5 ResNet-50**
> |CGMatch|Meta-Expert|CPG|SAGE|
> |---|---|---|---|
> |22.08|31.85|24.74|**35.93**|
>
> ## Scalability of Simplex Anchor Construction (W2)
>
> The QR and eigenvalue decompositions used for simplex anchor construction depend only on the backbone output dimension and are performed **once before training**, rather than during optimization. To assess scalability, we further evaluate SAGE with **ResNet-50 on CIFAR-10** (**2048**-dim features) and **HuBERT on ESC-50** (**748**-dim features). As shown in **Tables R5–R6**, SAGE still outperforms the strongest prior baseline by **4.08** and **2.00** points, respectively, demonstrating its practical effectiveness in higher-dimensional representation spaces.
>
> **Table R6 HuBERT**
> |FreeMatch|CGMatch|SimPro|CPG|SAGE|
> |---|---|---|---|---|
> |68.13|69.50|68.75|69.13|**71.50**|
>
> ## Adaptability to Different Numbers of Classes (Q1)
>
> The simplex anchors in SAGE are **not class prototypes**, so their number does **not need to coincide with the number of classes**. Instead, they define a fixed geometric coordinate frame for relational embedding. This is already reflected on **SVHN**, where the number of classes (**10**) is much smaller than the feature dimension (**128**): **Figs. 2(e–g) and 5** still show clearly improved feature separation.
>
> To further address this concern, we additionally vary the number of anchors $K$ on **CIFAR-10 with $(N=40,M=12390,\gamma_l=1,\gamma_u=100)$**. As shown in **Table R7**, performance changes only moderately across different choices of $K$ and remains consistently above competing methods, suggesting that SAGE is robust to the anchor number.
>
> **Table R7 Number of anchors $K$**
> |50|80|129 (used in paper)|200|
> |---|---|---|---|
> |65.30|63.98|66.83|63.87|
>
> Our total training time is 18.8 h vs. 18.5 h for FixMatch, indicating **only marginal overhead**, with anchor construction performed **only once before training (0.5 s, 0.0007% of total time)** and each use during training costing **only 0.007 s per iteration**. Please see our response to **Reviewer dZFX’s W3** for a detailed runtime analysis.
>
> ## Failure Boundary under Extreme Label Scarcity (Q2)
>
> We agree that a failure boundary may emerge under sufficiently extreme label scarcity. That said, our original paper already studies highly challenging UniSSL regimes with very few labels per class and shows clear gains. To further probe this boundary, we additionally evaluate SAGE on **CIFAR-10 with $(N=10,M=12390,\gamma_l=1,\gamma_u=100)$**, i.e., **1 labeled sample per class**. As shown in **Table R8**, SAGE still outperforms the strongest prior baseline by **5.71 points**, suggesting that relation-based structural inference remains effective even in this more extreme regime.
>
> **Table R8 Extreme Label Scarcity**
> |CGMatch|Meta-Expert|CPG|SAGE|
> |---|---|---|---|
> |37.31|17.84|32.12|**43.02**|
>
> ## Clarification on the Selection of Baselines in Table 3 (Q3)
>
> Due to space limits, **Table 3** includes a selected set of **recent, strong, and representative baselines**, including methods from **CVPR 2025, ICML 2025, and NeurIPS 2025**. Although the table is not exhaustive, it already covers the most competitive recent approaches. Even under this compact comparison, SAGE still achieves the best average accuracy and shows a clear gain over the strongest prior baseline.
>
> ## Clarification of the Introduction Statement on Relation Robustness (Q4)
>
> This statement is based on two empirical observations that were not stated clearly enough in the Introduction. **Fig. 2** shows that low-quality pseudo-labels can cause severe representation confusion in UniSSL, while **Fig. 3** shows that inter-sample relations can progressively correct erroneous pseudo-labels during training and eventually **stabilize at a high level**. We will revise the Introduction to make this motivation more explicit and directly connect it to these observations.
>
> ## Limitations and Future Directions (L1)
>
> We will revise the **Conclusion** to explicitly discuss the limitations of SAGE and outline possible future directions.

---

> > ### Author Rebuttal · Reviewer_Af3X · 2026-04-01
> >
> > Thank you for the detailed response; it has addressed most of my concerns, and I will maintain my positive rating. Meanwhile, I have two concerns. First, while the author emphasized the low overhead of anchor generation, I still doubt the scalability of this method on large-scale industrial datasets. The $O(M^2)$ complexity of the affinity matrix and the graph-state diffusion (whole-graph updating) may cost far beyond a marginal level, whereas baselines like FixMatch can be easily accelerated by reducing epochs or through parallelization. Second, the rigid isometric geometry may exert adverse effects on long-tailed distributions, potentially pulling tail data toward head-class anchors. I hope the author can further explain the theoretical boundaries or potential failure modes under these complex and imbalanced distributions.

---

> > > ### Author Response · Authors · 2026-04-02
> > >
> > > Thank you for your careful reading and valuable comments.
> > >
> > > ## Computational Overhead
> > >
> > > On the CIFAR-10 dataset, our method introduces only limited additional computational cost and remains comparable to recent baselines in terms of training time. On the same RTX 4090 GPU, the total training time is **18.5 h** for FixMatch, **29.5 h** for CGMatch, **19.2 h** for MetaExpert, **22.6 h** for CPG, and **18.8 h** for our method. Moreover, in most cases, the best performance is already achieved within the first **50%** of the total training time.
> > >
> > > The additional overhead is small for two main reasons.
> > >
> > > **First**, the anchor set is very small, with the number of anchors set to the output representation dimension plus one. In our default setting, we use only **129 anchors** for a **128-dimensional representation**. Therefore, the anchor-related computation is lightweight by design. Moreover, these anchors are fixed after construction rather than learned during training. They are generated only once before training, taking just **0.5 s** (**0.0007%** of the total training time), and introduce no iterative updates or additional optimization cost.
> > >
> > > **Second**, the additional computation during training is also lightweight. The main extra cost comes from the **batch-level relational inference** in GRI, which is **performed within each mini-batch rather than over the full unlabeled dataset**. In our setting, this reduces to a single matrix multiplication between a **448 × 128** matrix and a **128 × 129** matrix. The total runtime of the GRI module is only **0.007 s** per iteration. Since this operation involves only mini-batch features and a small fixed anchor set, its practical overhead is very limited.
> > >
> > > Overall, both anchor construction and relational inference are efficient, making the runtime of our method comparable to that of recent baselines. Specifically, when the output representation dimension is **128** and the batch size is **448**, the time required for anchor construction and relational inference is consistently about **0.5 s** and **0.007 s**, respectively, **regardless of the dataset**.
> > >
> > > ## Failure Boundary
> > >
> > > We agree that, under extreme long-tailed imbalance, fixed anchors may fail when the local structure of tail classes is too weak. However, in our method, the anchors are **not learned class prototypes but fixed, frequency-agnostic geometric references**, so they **do not inherently bias toward head classes**. This is also empirically supported by the discriminative representation shown in Fig. 2(g) under a severe imbalance ratio of 150, suggesting that the **proposed anchor mechanism remains robust even in highly imbalanced settings**. The real risk is instead that the structural consensus becomes dominated by dense head-class neighborhoods when tail samples are extremely sparse, class manifolds heavily overlap, or graph diffusion is too strong. This boundary is not yet formally characterized in the current paper, and we will clarify this limitation and discuss these failure modes explicitly in the revision.
> > >
> > > Thank you again for your valuable comments and positive rating. If you have any further concerns, please let us know. We would be more than happy to provide further clarification.

---

### Official Review · Reviewer_dZFX · 2026-03-09

**Soundness:** 4
**Presentation:** 3
**Significance:** 3
**Originality:** 3
**Overall Recommendation:** 4
**Confidence:** 4

**Summary:**

The manuscript introduces SAGE, a framework for Universal Semi-supervised Learning designed to handle arbitrary unlabeled data distributions under extreme label scarcity. By anchoring representations using fixed geometric structures inspired by neural collapse, SAGE creates a stable coordinate frame that bypasses the need for distribution priors. The framework employs three core strategies: a relational inference module to mine sample dependencies via graph diffusion, a reliability prioritization module that adaptively weights pseudo-labels, and an auxiliary head to isolate noisy gradients. By combining supervised, contrastive, and consistency losses, SAGE achieves state-of-the-art performance across multiple benchmarks, significantly outperforming existing semi-supervised and long-tailed learning methods.

**Compliance With Llm Reviewing Policy:**

Affirmed.

**Final Justification:**

The authors have addressed most of my concerns, so I will keep my positive score.

**Key Questions For Authors:**

Is the method scalable and capable of handling large or complex datasets? In addition, how to handle these four loss terms for a better trade-off?

**Limitations:**

yes

**Strengths And Weaknesses:**

Strengths:
- The core idea of shifting the paradigm from fragile distribution estimation to robust representation-level structural inference is compelling. Anchoring the representation space with fixed, geometrically optimal simplex equiangular frames provides a stable, distribution-agnostic foundation for learning.
- SAGE demonstrates significant and consistent improvements

Weaknesses:
- Although five datasets are used, they are all relatively small-scale image benchmarks. The paper does not demonstrate the method's effectiveness on larger-scale, more complex datasets (e.g., ImageNet)
- There are 4 loss terms, without any parameters to trade off each effect. It is suggested to fully explore whether we need some parameters for these terms and how to select them.
- GRI module involves constructing an affinity matrix and performing matrix exponentiation to derive the structural consensus. For a large unlabeled batch, computing A is computationally expensive. The manuscript does not discuss scalability or runtime compared to simpler baselines such as FixMatch.

---

> ### Author Rebuttal · Authors · 2026-03-30
>
> We appreciate the reviewer’s insightful comments and respond to each point below.
>
> ## Large or Complex Datasets (W1&Q1.1)
>
> We further evaluate SAGE on **ImageNet with WRN-28-2**. As shown in **Table R4**, SAGE achieves **53.83% accuracy**, outperforming the previous best baseline (**CPG, 50.43%**) by **3.40 points**, demonstrating its scalability to **larger-scale and more complex datasets**.
>
> **Table R4 ImageNet**
> |ACR|SimPro|CDMAD|Meta-Expert|CPG|SAGE|
> |---|---|---|---|---|---|
> |47.61|46.93|21.92|49.83|50.43|**53.83**|
>
> ## Parameters for Loss Terms (W2&Q1.2)
>
> We set all four loss weights to **1** in all experiments, without introducing extra balancing coefficients. Even without tuning these trade-off parameters, SAGE **consistently outperforms state-of-the-art methods**, with an average gain of **8.52 points** across five benchmark datasets. This suggests that SAGE achieves a good trade-off among the loss terms without additional weighting hyperparameters.
>
> ## Scalability or Runtime (W3)
>
> Our method introduces only limited additional computational cost and remains in the same training-time regime as recent baselines. On the same RTX 4090 GPU, the total training time is **18.5 h** for FixMatch, **29.5 h** for CGMatch, **19.2 h** for MetaExpert, **22.6 h** for CPG, and **18.8 h** for our method.
>
> The overhead is small mainly for two reasons. **First**, the anchor set itself is very small. In our default setting, we use only **129 anchors** for a **128-dim** representation, which is the minimal number needed to construct the simplex equiangular coordinate frame used in SAGE. Therefore, the anchor-related computation is lightweight by design. Moreover, these anchors are fixed after construction rather than learned during training. They are generated only once before training, taking just **0.5 s** (**0.0007%** of total training time), and introduce no iterative updates or additional optimization cost. **Second**, the additional computation during training is also lightweight. The main extra cost comes from **batch-level relational inference in GRI**, which is performed within each mini-batch rather than over the full unlabeled dataset. In our setting, it reduces to a single matrix multiplication of size **448×128 by 128×129**, taking only **0.007 s** per iteration. Since this operation involves only the mini-batch features and a small fixed anchor set, its practical overhead is very limited.
>
> Overall, both the anchor construction and the relational inference are efficient, so the runtime remains comparable to recent methods.

---

> > ### Author Rebuttal · Reviewer_dZFX · 2026-04-01
> >
> > The authors have addressed most of my concerns. However, simply setting all the weights to 1 for each loss term remains questionable. More sensitivity analysis is questioned. Hence, I will keep the same score.

---

> > > ### Author Response · Authors · 2026-04-02
> > >
> > > Thank you for your careful reading and valuable comments.
> > >
> > > Our loss function consists of two parts. The first part includes the supervised and unsupervised losses inherited from FixMatch, i.e., **L_cls** and **L_aux**. Their weights are fixed at 1.0, which is a standard setting commonly adopted in this line of work, and we follow this practice for a fair comparison. The second part includes the representation learning losses introduced by our GRI module, namely **L_con** and **L_sim**.
> > >
> > > To further address the reviewer’s concern, we conducted a sensitivity analysis on the weights of **L_con** and **L_sim** on CIFAR-10 under the setting $(N=40,M=12390,\gamma_l=1,\gamma_u=100)$. Specifically, we varied their shared weight from 0.1 to 3.0, and the results are reported in **Table R10**.
> > >
> > > The results show that the performance is relatively stable when the weight is set within a moderate range (e.g., 0.1 to 1.5), and the best performance is achieved around 1.0–1.5. In particular, in all experiments in the original paper, we use **a fixed default value of 1.0 for this hyperparameter without dataset-specific tuning**. Even so, our method still outperforms the competing methods by an average margin of **8.52%**, which shows that it does not depend on extensive hyperparameter tuning and remains efficient and practical in real applications.
> > >
> > > **Table R10 Sensitivity to Loss Weights**
> > >
> > > |Weight|0.1|0.4|1.0 (used in the paper)|1.5|2.0|2.5|3.0|
> > > | --- | --- | --- | --- | --- | --- | --- | --- |
> > > |Accuracy (%)|64.38|65.11|66.83|66.95|63.01|62.55|62.71|
> > >
> > > Thank you again for your valuable comments and positive rating. If you have any further concerns, please let us know. We would be more than happy to provide further clarification.

---

### Official Review · Reviewer_Hxnf · 2026-03-10

**Soundness:** 3
**Presentation:** 3
**Significance:** 3
**Originality:** 2
**Overall Recommendation:** 5
**Confidence:** 3

**Summary:**

This paper studies semi-supervised learning under extreme label scarcity and unknown unlabeled data distributions. The authors introduce a new setting called Universal Semi-supervised Learning (UniSSL), which assumes minimal labeled data and unlabeled data with arbitrary distributions.
To address this setting, the paper proposes SAGE (Simplex Anchored Graph-state Equipartition), which focuses on inferring structural relationships in the representation space rather than estimating the unlabeled data distribution. The framework combines three main components: (1) Graph-state Relational Inference (GRI) to capture higher-order sample relations, (2) simplex equiangular tight frame anchors to stabilize class representations, and (3) Distribution-agnostic Reliability Prioritization (DRP) to weight pseudo-labels based on confidence and margin without relying on distribution assumptions.

**Compliance With Llm Reviewing Policy:**

Affirmed.

**Final Justification:**

The responses of the authors have resolved most of my confusion, so I raise my score (3 -> 5).

**Key Questions For Authors:**

1.  What are the computational and memory costs of the graph-state relational inference module, especially for larger batch sizes or datasets?

2. Which components (GRI, anchors, DRP, auxiliary head) contribute most to the observed performance gains?

3. How sensitive is the method to representation dimensionality and the design of simplex anchors?

**Limitations:**

The paper could further discuss limitations such as the computational overhead of relational inference, sensitivity to hyperparameters, and scalability to larger datasets or architectures.

**Strengths And Weaknesses:**

### Strengths


1. The proposed framework is technically reasonable and supported by empirical results on several benchmark datasets. The paper includes ablation studies and representation analyses that provide additional evidence for the effectiveness of the proposed components.

2. The paper is generally well structured and the overall pipeline is clearly described. The motivation and intuition behind the approach are relatively easy to follow.

3. The paper addresses a practical semi-supervised learning scenario where labeled data are extremely limited and the unlabeled data distribution is uncertain, which is relevant for real-world applications.


4. The work combines relational inference, simplex anchor geometry, and pseudo-label reliability weighting into a unified framework. While individual ideas relate to prior work, their integration for distribution-agnostic semi-supervised learning provides a reasonably novel perspective.

### Weaknesses


1. The framework contains multiple components, and it is not entirely clear which part contributes most to the performance improvements. The computational cost of the graph-based relational inference module is also not clearly discussed.


2. Some technical details, such as the anchor construction and graph propagation process, could be explained more clearly.


3. Many components build on existing ideas in representation learning and semi-supervised learning, and the novelty mainly comes from their combination rather than entirely new techniques.

---

> ### Author Rebuttal · Authors · 2026-03-30
>
> We appreciate the reviewer’s insightful comments and respond to each point below.
>
> ## Component Contributions (W1.1&Q2)
>
> We analyze component contributions in two aspects. **First, all modules yield cumulative gains in overall performance**: **Table 4** shows that average accuracy improves from **45.56** to **49.58** (AB), **55.01** (DRP), and **57.89** (GRI). **Second, the main gains in pseudo-label quality come from DRP and GRI**: on CIFAR-10 with $(N=40,M=12390,\gamma_l=1,\gamma_u=100)$, pseudo-label F1 increases from **39.83** to **41.13** (AB), **45.58** (DRP), and **49.48** (GRI). **AB gives an initial boost, DRP improves pseudo-label reliability, and GRI further refines it through structural consensus.** We will include these results in the revised paper.
>
> ## Computational Overhead (W1.2&Q1&L1.1)
>
> Our method introduces limited overhead and has training time comparable to recent baselines. On the same RTX 4090 GPU, total training time is **18.5 h** for FixMatch, **29.5 h** for CGMatch, **19.2 h** for MetaExpert, **22.6 h** for CPG, and **18.8 h** for ours. Anchors are built only once before training, costing **0.5 s** (**0.0007%** of total time). During training, anchor-related computation is mainly one matrix multiplication (448×128 by 128×129), costing only **0.007 s** per iteration.
>
> ## Technical Clarifications (W2)
>
> The anchors in SAGE are **fixed simplex equiangular anchors** generated once before training, rather than learnable prototypes. Following Eq. (5), we obtain an orthogonal basis by QR decomposition, apply the centering matrix to form a zero-centered simplex frame, and then construct the anchors. Thus, they are **zero-centered, unit-norm, and pairwise equiangular**, providing a stable coordinate frame.
>
> For graph propagation, each unlabeled sample is first represented in the anchor coordinate frame to encode its relations to all anchors. Pairwise similarities are then computed in this relational space to build a graph, whose rows are normalized by softmax into transition probabilities, followed by **multi-step propagation**. This lets each sample aggregate information from **immediate and higher-order neighbors**, yielding a more reliable structural signal.
>
> We will clarify the anchor construction and graph propagation process in the final version.
>
> ## Design Motivation and Intuition (W3)
>
> Our contribution is **not a simple combination of existing ideas**, but a targeted solution to a key failure mode in **UniSSL**: under extreme label scarcity and unknown unlabeled data distributions, pseudo-labels become unreliable and existing methods break down. Prior works [1,2] show that **self-supervised representation learning** learns robust, invariant features. We further observe that in UniSSL, **inter-sample relations are more reliable than pseudo-labels**. We therefore shift the learning target from **pseudo-label supervision and distribution estimation** to **representation-level structural consensus**, leading to more robust learning and better performance in UniSSL.
>
> [1] NeurIPS’22-Contrastive and Non-Contrastive Self-Supervised Learning Recover Global and Local Spectral Embedding Methods
>
> [2] CVPR’21-Understanding the Behaviour of Contrastive Loss
>
> ## Sensitivity to Representation Dimensionality and Anchor Design (Q3)
>
> In the main paper, we use **WRN-28-2** with a **128-dim** output feature. We further evaluate SAGE with **ResNet-50** and a **2048-dim** output feature. **Table R1** shows that it still outperforms the strongest prior baseline by **4.08 points**, showing effectiveness with a different representation dimensionality. We also vary the **number of anchors $K$**. **Table R2** shows only moderate variation across different $K$ values, while performance remains consistently above competing methods, suggesting robustness to anchor design. All experiments are on CIFAR-10 with $(N=40,M=12390,\gamma_l=1,\gamma_u=100)$.
>
> **Table R1 ResNet-50**
> |CGMatch|Meta-Expert|CPG|SAGE|
> |---|---|---|---|
> |22.08|31.85|24.74|**35.93**|
>
> **Table R2 Number of anchors $K$**
> |50|80|129 (used in paper)|200|
> |---|---|---|---|
> |65.30|63.98|66.83|63.87|
>
> ## Hyperparameter Sensitivity (L1.2)
>
> Our hyperparameters are **fixed across experiments**, rather than tuned per setting. **Appendix D / Fig. 6** shows that SAGE is **reasonably robust** to the two key hyperparameters, $\lambda$ and $\beta$.
>
> ## Larger Datasets or Architectures (L1.3)
>
> We further evaluate SAGE on **larger backbones and datasets**. With **ResNet-50 on CIFAR-10**, SAGE still outperforms the strongest prior baseline by **4.08 points** (**35.93** vs. **31.85**; **Table R1**). On **ImageNet with WRN-28-2**, it also surpasses the strongest prior baseline by **3.40 points** (**53.83** vs. **50.43**; **Table R3**). These results suggest that SAGE generalizes well to both **larger datasets and architectures**.
>
> **Table R3 ImageNet**
> |ACR|SimPro|CDMAD|Meta-Expert|CPG|SAGE|
> |---|---|---|---|---|---|
> |47.61|46.93|21.92|49.83|50.43|**53.83**|

---

> > ### Author Rebuttal · Reviewer_Hxnf · 2026-04-01
> >
> > Thank the authors for their responses, which resolved most of my confusion, so I raised my score (3 -> 5).

---

> > > ### Author Response · Authors · 2026-04-01
> > >
> > > Dear Reviewer **Hxnf**,
> > >
> > > We are glad that your concerns have been adequately addressed. Thanks for raising the score, and thanks again for your valuable comments and suggestions to improve this paper.
> > >
> > > Regards from the authors.

---

### Decision · Program_Chairs · 2026-04-30

**Decision:**

Accept (regular)

**Comment:**

This paper addresses Universal Semi-supervised Learning under extreme label scarcity and unknown unlabeled data distributions. Overall,  reviewers consistently noted the strength of the idea, and the method shows gains over prior baselines.
Several reviewers explicitly stated that their concerns were fully or largely addressed, and the final recommendations are positive overall. The remaining weaknesses mainly concern scalability in very large industrial settings, but these do not outweigh the paper’s merits.
I therefore lean toward acceptance.